# Rubella virus tropism and single-cell responses in human primary tissue and microglia-containing organoids

**Galina Popova**[1,2,3,4,5,6†], **Hanna Retallack**[6†], **Chang N Kim**[1,2,3,4,5,6], **Albert Wang**[1,2,3,4,5,6], **David Shin**[1,2,3,4,5,6], **Joseph L DeRisi**[6,7*], **Tomasz Nowakowski**[1,2,3,4,5,6*]

[1]Department of Neurological Surgery, University of California, San Francisco, San Francisco, United States; [2]Eli and Edythe Broad Center for Regeneration Medicine and Stem Cell Research, University of California, San Francisco, San Francisco, United States; [3]Department of Anatomy, University of California, San Francisco, San Francisco, United States; [4]Department of Psychiatry and Behavioral Sciences, University of California, San Francisco, San Francisco, United States; [5]Weill Institute for Neurosciences, University of California, San Francisco, San Francisco, United States; [6]Department of Biochemistry and Biophysics, University of California, San Francisco, San Francisco, United States; [7]Chan Zuckerberg Biohub, San Francisco, United States

**\*For correspondence:**
joseph.derisi@ucsf.edu (JLDeR);
tomasz.j.nowakowski@gmail.
com (TN)

†These authors contributed
equally to this work

**Competing interest:** The authors
declare that no competing
interests exist.

**Reviewing Editor:** Joseph G
Gleeson, University of California,
San Diego, United States

**Abstract** Rubella virus is an important human pathogen that can cause neurological deficits in a developing fetus when contracted during pregnancy. Despite successful vaccination programs in the Americas and many developed countries, rubella remains endemic in many regions worldwide and outbreaks occur wherever population immunity is insufficient. Intense interest since rubella virus was first isolated in 1962 has advanced our understanding of clinical outcomes after infection disrupts key processes of fetal neurodevelopment. Yet it is still largely unknown which cell types in the developing brain are targeted. We show that in human brain slices, rubella virus predominantly infects microglia. This infection occurs in a heterogeneous population but not in a highly microglia-enriched monoculture in the absence of other cell types. By using an organoid-microglia model, we further demonstrate that rubella virus infection leads to a profound interferon response in non-microglial cells, including neurons and neural progenitor cells, and this response is attenuated by the presence of microglia.

## eLife assessment

The manuscript represents an **important** study on the pathogenesis of rubella virus tropism and neuropathology in human microglia-containing human stem cell derived organoids and human fetal brain slices. The strength of evidence is **compelling**, employing two different human-relevant models. The findings will be of broad interest to virologists and infectious disease experts, as well as neurodevelopmental biologists. The findings could also be of interest to pediatrics and obstetrics clinical colleagues.

## Introduction

Neurotropic viruses contracted during pregnancy can have grave consequences for the fetus. These comprise both viruses of longstanding concern like human cytomegalovirus and herpes simplex virus as well as emerging viruses like Zika virus. Yet our understanding of how direct viral infection and

indirect inflammatory consequences affect fetal brain development is limited. This is true even for well-studied pathogens like rubella virus (RV), which is an enveloped, single-stranded RNA virus of the family Matonaviridae restricted to human transmission. Infection with RV typically causes a mild, self-limiting illness with a characteristic rash during childhood, often referred to as 'German measles'. However, infection during pregnancy can cause miscarriage, stillbirth, or a range of birth defects including congenital rubella syndrome (CRS). The sequelae of congenital RV infection were first recognized in 1941 and although the first RV vaccines were licensed in 1969, an estimated 105,000 infants with CRS were born each year worldwide as of 2010 (*Vynnycky et al., 2016*). As of 2019, RV-containing vaccine coverage remains incomplete and inconsistent, with ongoing endemic transmission and reporting gaps primarily in the African, Eastern Mediterranean, and South-East Asian World Health Organization Regions (*World Health Organization, 2020*). Countries with RV-containing vaccine programs also remain susceptible to outbreaks, such as Japan and China, where outbreaks in 2013–14 and 2018–19 caused a twofold increase in reported rubella cases worldwide (26,033 total cases in 2018 vs 49,179 cases in 2019) (*Plotkin, 2021*) and included CRS the following year (423 total cases worldwide in 2019 vs 1252 cases in 2020) (*World Health Organization, 2022*).

The most common features of CRS are congenital cataracts, sensorineural deafness, and cardiac defects (*Banatvala and Brown, 2004*). In addition, microcephaly (*Munro et al., 1987*), developmental delay and autism (*Chess, 1977*), and schizophrenia spectrum disorders (*Brown et al., 2001*) are associated with the syndrome, but the pathophysiology of these neurological complications is not well described. To gain mechanistic insight into the pathophysiology of CRS, it is essential to understand the tropism of the virus. Initial infection in the lymphoid tissues of the nasopharynx and upper respiratory tract leads to systemic viremia, with virus spread across the placenta and into nearly all fetal organs on post-mortem examination, primarily via infected mononuclear cells (*Nguyen et al., 2015*). As for the fetal nervous system, RV was isolated from cerebrospinal fluid and brain tissue of fetuses and infants with CRS in studies from the 1960s (*Bellanti et al., 1965*; *Esterly and Oppenheimer, 1967*; *Korones, 1965*; *Monif et al., 1965*). However, further details of where RV might replicate in the brain are lacking. Autopsies in that early era revealed nonspecific gliosis and cerebral vessel degeneration (*Rorke and Spiro, 1967*). In limited pathology specimens from more recent outbreaks, RV RNA and antigens were identified in rare cells in the cortex and cerebellum presumed to be 'nerve cells' and neural progenitor cells (*Lazar et al., 2016*; *Nguyen et al., 2015*). Experimental infections of cells that might not accurately represent the primary cells in the developing brain have yielded little further insight. To complicate the matter, myelin oligodendrocyte glycoprotein (MOG) has been proposed as a cellular receptor for RV (*Cong et al., 2011*), but it is exclusively expressed in oligodendrocytes in the human brain and therefore cannot explain infection in other cell populations. Thus, there is clear evidence for the presence of RV in the central nervous system in infants with CRS, but the identity of infected cell type(s) remains elusive.

Here, we address RV tropism in the human developing brain and other poorly understood molecular aspects of CRS. By combining primary human brain tissue with a variety of cell culture techniques, we show that microglia are the predominant cell type infected by RV. Furthermore, we show that diffusible factors from non-microglia cells are necessary to render microglia susceptible to RV. By using brain organoids supplemented with primary microglia, we demonstrate that RV infection leads to a robust interferon response and leads to dysregulation of multiple genes implicated in human brain development. Finally, we compared transcriptomic changes between microglia-transplanted and non-transplanted organoids and found that in the presence of microglia, interferon pathway upregulation following RV exposure is reduced.

## Results
### RV infects microglia in the human developing brain
To investigate RV tropism in the human brain, cultured cortical slices from mid-gestation samples were infected with M33 RV, representing a laboratory strain originally derived from a clinical isolate (*Figure 1A*). At 72 hr post-infection, immunostaining for the RV capsid protein revealed numerous cells positive for the RV antigen, of which >90% were co-labeled with the microglia marker Iba1 (*Figure 1B–D*). To confirm functional transcription and translation of the viral genome, a new reporter construct of RV designed to express GFP within the non-structural P150 gene was generated (RV-GFP,

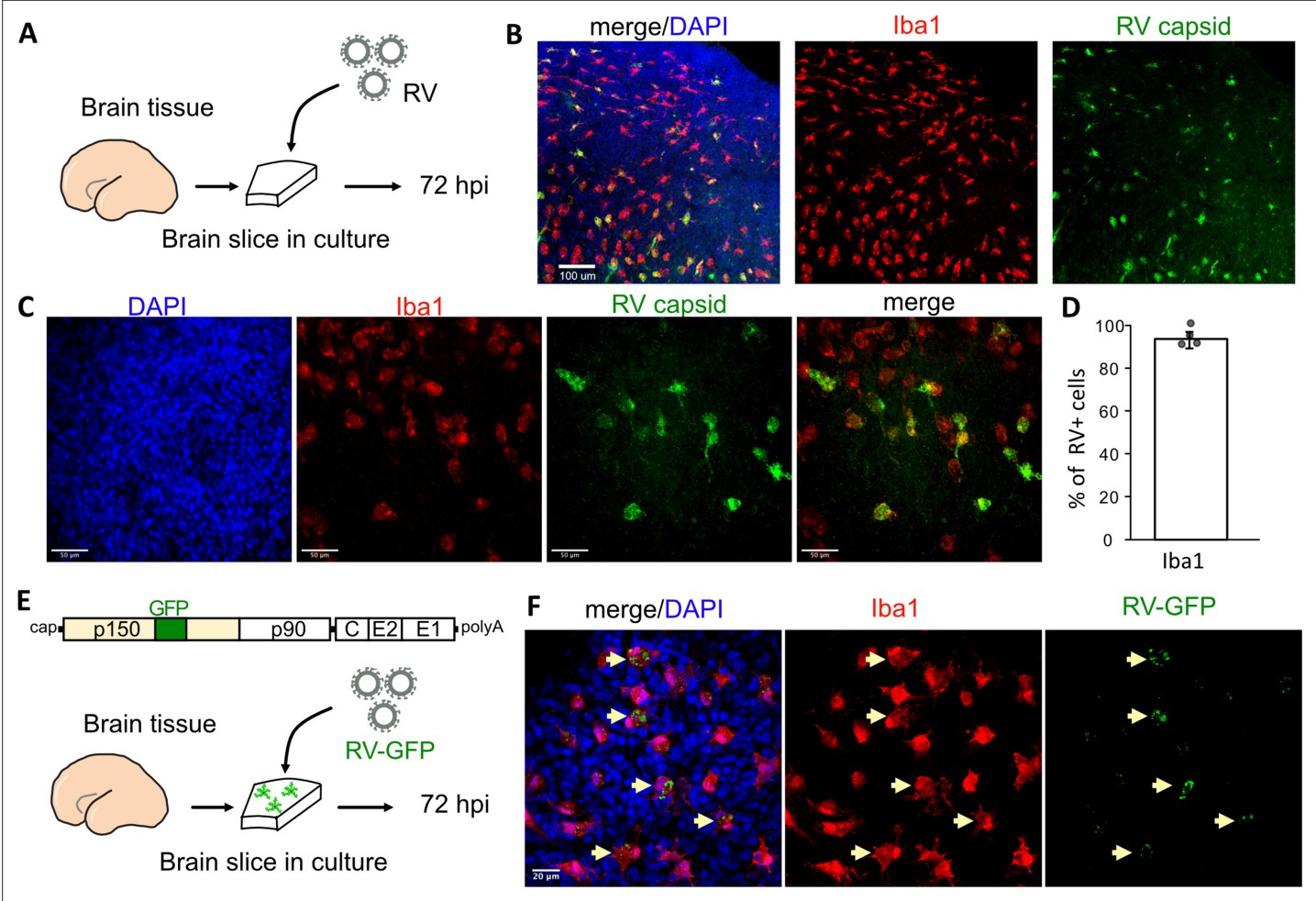

**Figure 1.** Rubella virus (RV) infects primary human microglia in cultured brain slices. (**A**) Schematic for brain slice infection. Mid-gestation (GW18-23) human brain slices were infected with RV for 72 hr. (**B, C**) Immunostaining for RV capsid and Iba1 in cultured cortical slices at 72 hpi, at 10× (scale bar 100 μm) (**B**) and at 40× magnification (scale bar 50 μm) (**C**). (**D**) Quantification of RV capsid-positive cells co-labeled with microglial marker Iba1: 764/819 (93.3%) of RV+ cells were microglia based on Iba1 staining across four biological replicates. Error bars denote standard deviation. (**E**) Diagram of viral genome of GFP-expressing RV (RV-GFP). Cortical brain slices were infected with RV-GFP for 72 hr. (**F**) Examples of GFP fluorescence and Iba1 immunostaining at 72 hpi of cultured cortical slices with GFP-RV, at 62× (scale bar 20 μm). GFP expression of modified RV is localized to Iba1-positive microglia cells (arrows).

The online version of this article includes the following figure supplement(s) for figure 1:

**Figure supplement 1.** GFP expression in rubella virus (RV)-infected Vero cells.

GenBank Accession OM816675, *Figure 1E*) and validated by GFP expression in Vero cells (*Figure 1— figure supplement 1*). In human primary brain slices infected with RV-GFP, GFP expression was detected predominantly in microglia, confirming the production of RV proteins inside this cell type (*Figure 1F*) consistent with the wild type M33 RV.

## Cell microenvironment influences RV infectivity

Such specificity of RV for microglia in this model is striking given that microglia represent only 1–5% of the cells of the human developing brain (*Menassa et al., 2022*). Moreover, the previously published viral entry factor MOG is not specific to microglia according to analysis of publicly available RNA and protein expression profiles of the human developing brain (*Nowakowski et al., 2017*) (https://cells. ucsc.edu/?ds=cortex-dev&gene=MOG) or human radial glial cells (*Eze et al., 2021*) (https://cells. ucsc.edu/?ds=early-brain&gene=MOG). Further, common components of the host cell membrane, such as sphingomyelin and cholesterol that appear to be essential for RV entry (*Otsuki et al., 2018*), cannot explain viral tropism for microglia. Thus, to identify factors contributing to the relatively

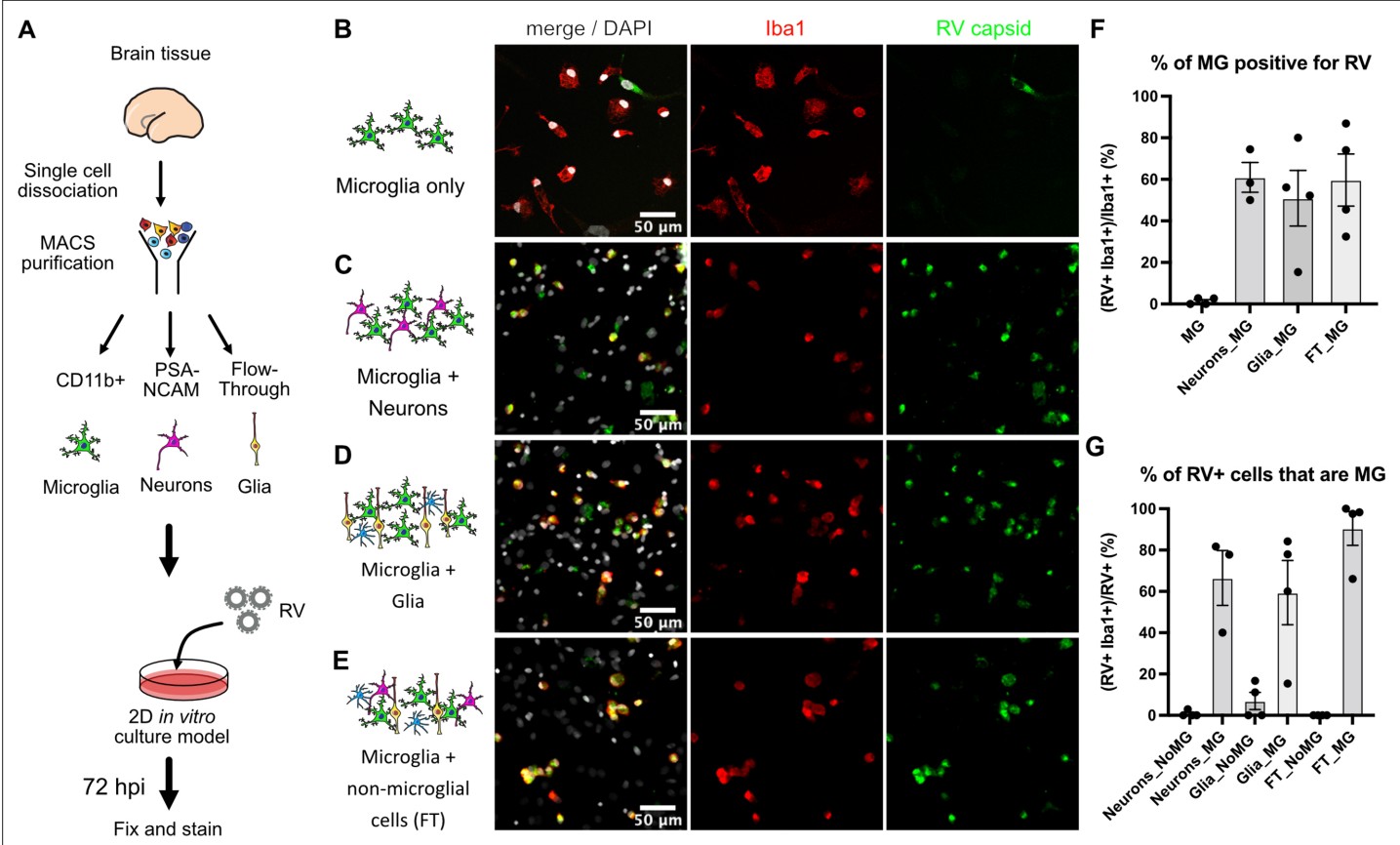

**Figure 2.** Rubella virus (RV) infection of microglia is dependent on the presence of other cells. (**A**) Schematic of rubella infection. Primary prenatal brain tissue was dissociated and different cell types were purified using magnetic-activated cell sorting (MACS). Microglia cells were cultured alone or in combination with neurons, glial cells, or all cell types. 2D cultures were infected with RV for 72 hr and processed for immunostaining. (**B–E**) Representative images of microglia cultured with different cell types. Cell cultures were stained for microglia marker Iba1 (red), RV capsid (green), and DAPI (gray; on the overlay Merge channel). (**B**) Purified microglia only. (**C**) Microglia and neurons (purified with PSA-NCAM magnetic beads) co-cultured at 1:5 ratio. (**D**) Microglia and glial cell types (flow-through fraction after PSA-NCAM magnetic beads) cultured together at 1:5 ratio. (**E**) Microglia cultured with non-microglial cells (flow-through after CD11b magnetic beads; mixed cell populations) at 1:5 ratio. (**F**) Quantification of RV capsid immunopositivity among microglia (Iba1+) for conditions in B–E. FT: flow-through after microglia MACS purification. Error bars denote SEM. Each data point (N=4) represents a field of view from the same experimental batch and represents a technical replicate. (**G**) Quantification of microglia (Iba1+) among RV capsid-positive cells.

The online version of this article includes the following figure supplement(s) for figure 2:

**Figure supplement 1.** Rubella virus (RV) inoculum dilution in mixed co-cultures of microglia and non-microglia cells.

**Figure supplement 2.** Rubella infection in non-microglia cells.

specific infection of microglia, RV infectivity was tested in monocultures of primary human microglia. Microglia from mid-gestation cortical brain samples were purified using magnetic-activated cell sorting (MACS) and then subsequently infected with RV (**Figure 2A**). Surprisingly, RV infection of the microglia monoculture was negligible (**Figure 2B**). To resolve this apparent paradox, we investigated whether microglia infectivity could be restored by the presence of other cell types, such as neurons or progenitor cells. Microglia were co-cultured with either neuronally enriched cultures (sorted with PSA-NCAM magnetic beads) or the glial component (flow-through that was depleted of both the CD11b-positive microglia cells and the PSA-NCAM-positive population). Both conditions together with mixed brain cells (flow-through from CD11b-depleted fraction; FT) successfully restored infection (**Figure 2C–E**). In the pure microglial cultures, less than 2% of microglia were positive for RV capsid by immunostaining, but when different cell fractions were added to the culture (neuronal, glial, or mixed cultures), up to 60% of microglia had RV capsid immunopositivity (**Figure 2F**). Similar to the cortical brain slices, microglia represented the main cell type infected with RV in the mixed co-cultures

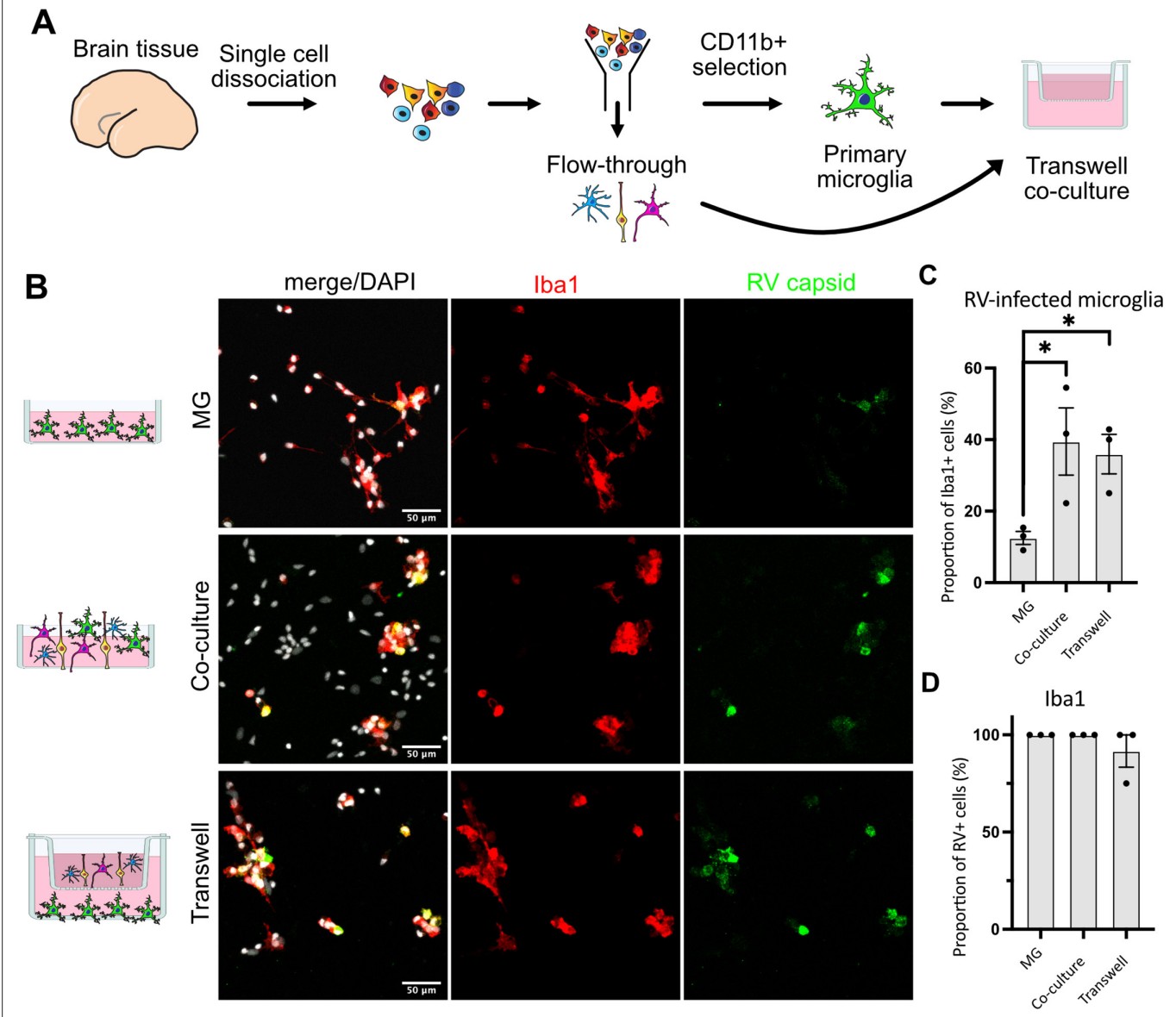

**Figure 3.** Direct cell-cell contact is not required for microglia infection by rubella virus (RV). (**A**) Schematic for experimental setup. Primary human brain tissue was dissociated, and microglia were cultured with or without microglia-depleted flow-through portion. Cells were co-cultured in direct contact or in solution-permeable chambered transwells (TW). (**B**) Representative images of microglia-enriched cultures (top row), microglia cultured with other cell types in the same well (middle row), and microglia cultured in the bottom compartment with other cell types cultured in a permeable transwell chamber (bottom row) infected with RV for 72 hr. (**C**) Quantification of RV capsid immunopositivity among microglia (Iba1+). Three fields of view across the same experiment were quantified for each condition and represent technical replicates. Error bars represent SEM. p-value between microglia and co-culture condition is 0.0479. p-value between microglia and transwell condition is 0.0159. (**D**) Quantification of microglia (Iba1+) among RV capsid-positive cells.

(*Figure 2G*). Furthermore, mixed cultures inoculated with lower viral titers had fewer cells with RV capsid immunopositivity overall, but retained a high proportion of infected microglia demonstrating specificity for microglia (*Figure 2—figure supplement 1A–C*). Despite RV capsid co-localizing with microglia cells and GFP protein being produced from RV-GFP in microglia, RV titering experiments failed to detect significant production of newly released virions in microglia co-cultures (*Figure 2— figure supplement 1D*). No statistically significant difference was detected in RV infectivity in cells cultured with or without microglia (*Figure 2—figure supplement 2*).

We then tested whether RV capsid immunopositivity in microglia could be due to phagocytic activity by this macrophage population. To exclude microglia engulfing other infected cells, a transwell system

was employed where microglia and other cell types are grown in compartments separated by a semi-permeable membrane that allows media exchange without direct cell-cell contacts (*Figure 3A*). Both the presence of other cell types in the same well (co-culture) and the media exchange between the two chambers (transwell) restored infection in microglia (*Figure 3B–C*). Consistent with our previous experiments, microglia represented the main cell type infected with RV (*Figure 3D*). Together, these results suggest that RV capsid immunostaining cannot be explained by phagocytosis of other infected cells, but it is possible that infection of microglia is influenced by diffusible factors from other cell populations found in the tissue microenvironment.

## Rubella infection elicits an interferon response in brain organoids

Given the striking difference in infection rates in different cell environments, we next investigated how the presence of microglia modulates response to the viral infection in other cell populations. We used brain organoids as a model of early brain development that, unlike primary brain slices, can be cultured for prolonged period of time to investigate long-term consequences of RV exposure. Under standard protocols, brain organoids do not robustly develop any cells of myeloid origin, making them a useful reductionist model for investigating the role of immune cells in brain homeostasis and development (*Nowakowski and Salama, 2022*). Brain organoids were generated following previously established protocols (*Paşca et al., 2015*), and at 5 weeks of differentiation, when the majority of cell types are present in the organoids, mid-gestation primary human microglia were introduced as previously described (*Popova et al., 2021*). After allowing microglia to engraft into the organoids, we exposed neuroimmune organoids to RV or heat-inactivated control and cultured them for 72 hr or 2 weeks to identify short- and long-term consequences of the viral exposure (*Figure 4A*). In organoids with engrafted primary microglia subsequently exposed to RV, immunostaining revealed RV capsid in microglia, similar to primary tissue and co-culture experiments. We detected RV capsid in microglia, but not in other cell types, in both timepoints, suggesting that microglia remain the main cell population that harbors viral infection (*Figure 4B–C*).

To determine brain-wide consequences of the RV infection across different cell types, at 72 hr after RV exposure we processed neuroimmune organoids for single-cell RNA sequencing (scRNAseq) with 10x Genomics and downstream analysis. After processing for scRNAseq, cells with fewer than 500 detected genes and/or more than 20% mitochondrial genes were removed from the analysis. Ribosomal transcripts and pseudogenes were excluded. Approximately 11,000 cells passed filtering criteria (*Figure 5A*, *Figure 5—figure supplement 1A–C*), revealing the expected major cell populations of the human developing brain (*Nowakowski et al., 2017*), including radial glia cells, immature and mature neurons, and astrocytes (*Figure 5A–B*). Cell cluster annotations were assigned based on combinations of co-expressed cluster marker genes, such as *FGFBP2* and *SOX2* for radial glial cells (clusters 5 and 10), TAGLN3, *HES6*, *NEUROD4* for neural progenitor cells (cluster 7), *TUBB2A, TUBB2B, STMN2* for neurons (cluster 2), *CLU, PTN,* and *SPARCL1* for astrocytes (cluster 6), and *MKI67, UBE2C,* and *CENPF* for dividing cells (clusters 3 and 4) (*Figure 5B*, *Figure 5—figure supplement 1D–E* for individual cluster marker genes, *Figure 5—source data 1* for the full list of markers). Cells derived from organoids with and without microglia were present in all clusters (*Figure 5C*). A separate microglia cluster was not identified. Rare cells expressing the microglia marker *AIF1* (encoding the Iba1 protein) were present, but such cells have been previously reported to develop spontaneously in organoids (*He et al., 2022*) and the canonical microglia marker *P2RY12* was not detected in those cells (*Figure 5—figure supplement 1F*). We attribute the apparent lack of microglia to both the small starting population and loss due to cell dissociation during processing for scRNAseq. Consistent with the lack of microglia cells in our scRNAseq data, we did not recover appreciable numbers of the viral transcripts. However, exposure of organoids to RV resulted in significant transcriptomic differences including genes involved in the interferon signaling pathway and its response (*IFI27, IFI6, IFITM3*) (*HLA-A* [*Campbell et al., 1986*; *Keskinen et al., 1997*] and *BST2* [*Holmgren et al., 2015*]) (cluster 1, *Figure 5E*, *Figure 5—figure supplement 1* and *Figure 5—source data 1*). The majority of cells in cluster 1 came from RV-exposed organoids (*Figure 5D*). While genes involved in the interferon response showed increased expression in organoids both with and without microglia, the magnitude of their upregulation was lower among cells in microglia-containing organoids (*Figure 5E–F*). We confirmed higher expression levels of IFITM3 protein in organoids with microglia exposed to RV in comparison to organoids with microglia exposed to heat-inactivated RV

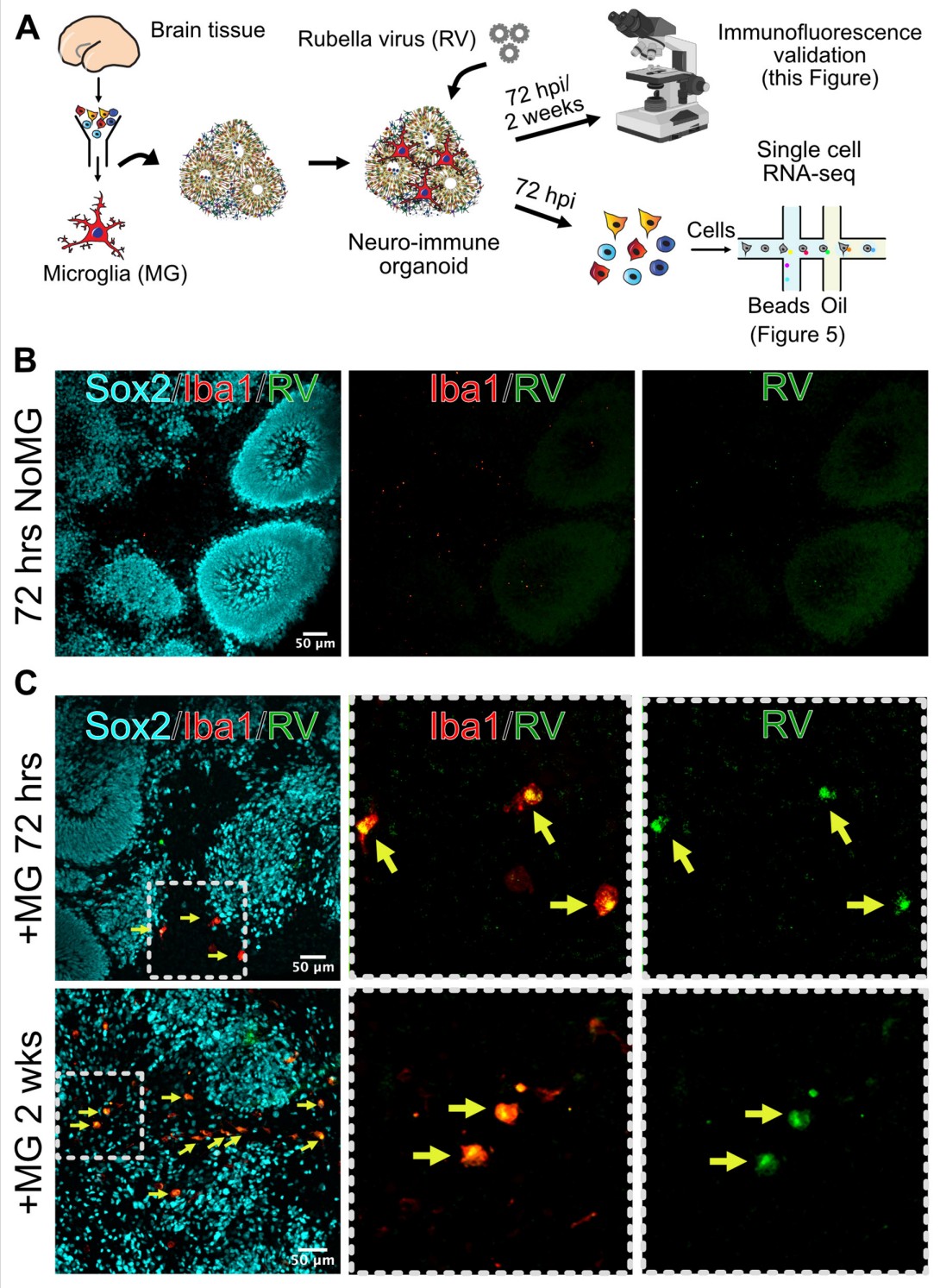

**Figure 4.** Microglia in neuroimmune organoids are infected with rubella virus (RV). (**A**) Primary human microglia were transplanted into brain organoids and resulting neuroimmune organoids were exposed to RV. After 72 hr or 2 weeks organoids were processed for immunofluorescence validation (this figure) or single-cell RNA sequencing (scRNAseq) analysis (***Figure 5***). (**B**) Representative immunofluorescence images of brain organoids without microglia subjected to RV exposure for 72 hr. Radial glial cells are labeled with Sox2 (cyan), microglia are labeled with Iba1 (red), and RV is labeled with anti-RV capsid antibody (green). Scale bar is 50 μm. (**C**) Representative immunofluorescence images of brain organoids with microglia at 72 hr (top panel) or 2 weeks (bottom panel) after RV exposure. Radial glial cells are labeled with Sox2 (cyan), microglia are labeled with Iba1 (red), and RV is labeled with anti-RV capsid antibody (green). Dashed boxes represent zoomed-in examples of microglia cells. Scale bar is 50 μm.

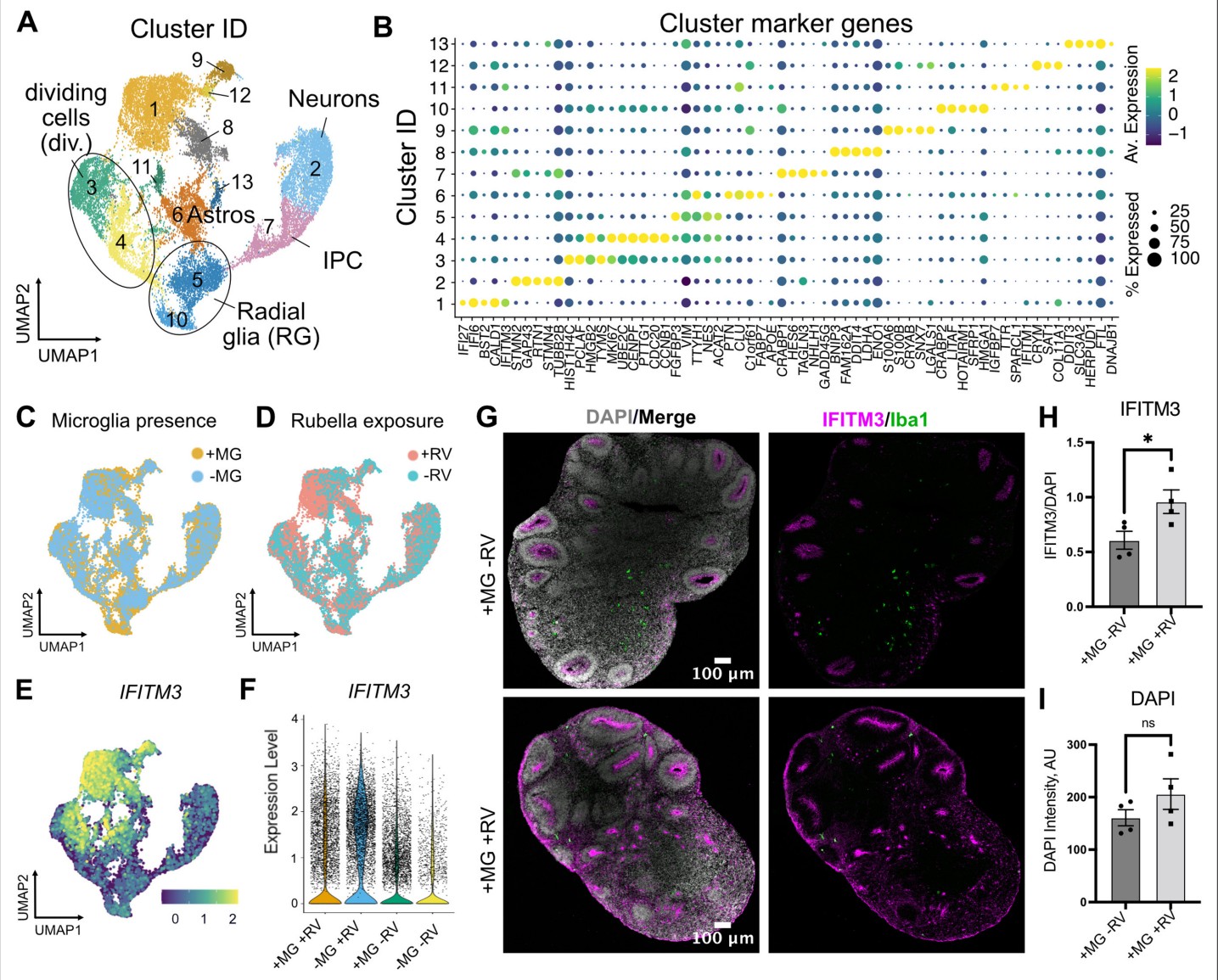

**Figure 5.** Rubella virus (RV) exposure of brain organoids leads to interferon response. (**A**) Single-cell RNA sequencing analysis identified 13 clusters, including neurons and glial cells (Div.: dividing cells, RG: radial glia, Astros: astrocytes, IPCs: intermediate progenitor cells). (**B**) Dot plot depicting cluster marker genes for each cluster. (**C**) Uniform manifold approximation and projection (UMAP) plots of organoids colored by the presence or absence of microglia. (**D**) UMAP plots of organoids colored by the presence or absence of RV treatment. (**E**) Feature plot for expression levels of *IFITM3*. (**F**) *IFITM3* expression in all cells across different conditions. (**G**) Representative images of *IFITM3* immunofluorescence in brain organoids with microglia with wild type RV (bottom panel) or heat-inactivated control (top panel) 72 hr post-infection. IFITM3 is labeled in magenta, microglia are labeled with Iba1 (green), cell nuclei are labeled with DAPI (gray). (**H**) Quantification of fluorescence intensity of IFITM3 normalized to DAPI intensity per organoid. Columns represent mean of four organoids. Dots represent averages across several sections for each individual organoid. Error bars represent SEM. Unpaired parametric Student's t-test was used to compare the two groups in H–I. p-value = 0.04. (**I**) Quantification of fluorescence intensity of DAPI staining per organoid. p-value = 0.22.

The online version of this article includes the following source data and figure supplement(s) for figure 5:

**Source data 1.** Cluster marker genes for brain organoid single-cell RNA sequencing (scRNAseq) dataset, related to *Figure 5*.

**Source data 2.** Differentially expressed genes detected in brain organoid single-cell RNA sequencing (scRNAseq) dataset, related to *Figure 5*.

**Figure supplement 1.** Single-cell RNA sequencing (scRNAseq) analysis of brain organoids.

at 72 hr post-exposure (*Figure 5G–H*), while the overall cell numbers were not changed in either condition (*Figure 5I*).

To investigate how the presence of microglia and RV exposure modulate gene expression profiles across different cell populations, we stratified gene expression differences based on the major cell types for each of the four conditions (absence and presence of microglia; absence or presence of RV exposure). Radial glia and dividing cells had a greater transcriptomic response to RV exposure as compared to neurons, both with and without microglia (*Figure 6A*). Cells captured from microglia-containing organoids showed fewer differentially expressed genes in response to RV in each of the major cell classes compared to organoids that did not contain microglia (*Figure 6A*, top vs bottom panel), with radial glia and neurons reaching statistically significant levels (p-values shown on the right side of the panel) and neural progenitor cells showing the overall trend without reaching statistical significance. One gene family that was specifically downregulated in the presence of RV in organoids without microglia included nuclear factor I – *NFIB* and *NFIA* (*Figure 6A*, *Figure 5—source data 2*) – two genes that form heterodimers in vivo and are associated with induction of gliogenesis (*Tchieu et al., 2019*) in embryonic brain development. Early disruption in the function of either gene is associated with neurodevelopmental deficits and perinatal mortality in mice (*das Neves et al., 1999*; *Steele-Perkins et al., 2005*) and with intellectual disability in humans (*Schanze et al., 2018*).

Genes with expression levels affected both by the presence of microglia and by RV exposure included NOVA alternative splicing regulator 1 (*NOVA1*) (*Figure 6B*). *NOVA1* is a master regulator of alternative splicing (*Zhang et al., 2010*) in the central nervous system with potential links to neurological diseases (*Parikshak et al., 2016*). Unlike primary brain slices, brain organoids can be cultured for extended periods of time, providing a human-specific model for studying long-term consequences of RV infection. To better mimic normal human brain development, we used neuroimmune organoids with microglia exposed to RV or heat-inactivated controls to determine how the presence of the viral infection influences *NOVA1* expression and neuronal cell differentiation. After 2 weeks, we used immunostaining to quantify numbers of neurons or intermediate progenitor cells (IPCs) – two major cell types with the most robust predicted NOVA1 level changes based on the scRNAseq experiment (*Figure 6C*). We detected a statistically significant decrease of NOVA1+ IPCs in response to RV exposure (*Figure 6E*). Numbers of NOVA1+ neurons also had a trend toward reduction (*Figure 6D*); however, it did not reach statistical significance. Concurrently with reduction of NOVA1+ cells, we detected lower numbers of neurons (*Figure 6F*), but not IPCs (*Figure 6G*), in organoids after RV exposure.

## Discussion

Here, we demonstrate that in the developing brain RV predominantly infects microglia, the resident macrophage population. This finding is consistent with RV tropism for monocytes in the periphery (*Perelygina et al., 2021*; *van der Logt et al., 1980*), and adds new information to the limited understanding of RV infection in the central nervous system. Supporting data from real-world infections including post-mortem specimens would be helpful to evaluate clinical strains. Tropism for microglia raises interesting questions about how and where RV persists in CRS, perhaps in brain tissue during the extended period of viral shedding, similar to other relatively immuno-privileged sites such as the eye (*Doan et al., 2016*; *Sugishita et al., 2016*). Notably, we did not detect significant production of newly released virions, suggesting potential limitations of the culture system. Our findings also help contextualize CRS in comparison to congenital infections by other neurotropic viruses: human immunodeficiency virus type 1 and Zika virus, which target microglia directly; herpes simplex virus, which replicates poorly in microglia with cytopathic effect; and human cytomegalovirus, which causes microglia to produce antiviral cytokines without productive infection or cytopathic effect (*Lum et al., 2017*; *Retallack et al., 2016*; *Rock et al., 2004*).

Like some of these other viruses, we found that by establishing viral transcription and translation in microglia, RV elicits a strong interferon response in other cell types. It has been previously shown that the interferon response in neurons derived from induced pluripotent stem cells can induce molecular and morphological changes associated with neurodevelopmental disorders, including neurite length and gene expression changes associated with schizophrenia and autism (*Warre-Cornish et al., 2020*). The interferon response is additionally associated with pathobiology in a range of congenital infections and interferonopathies (*Crow and Manel, 2015*). Furthermore, in our preliminary experiments

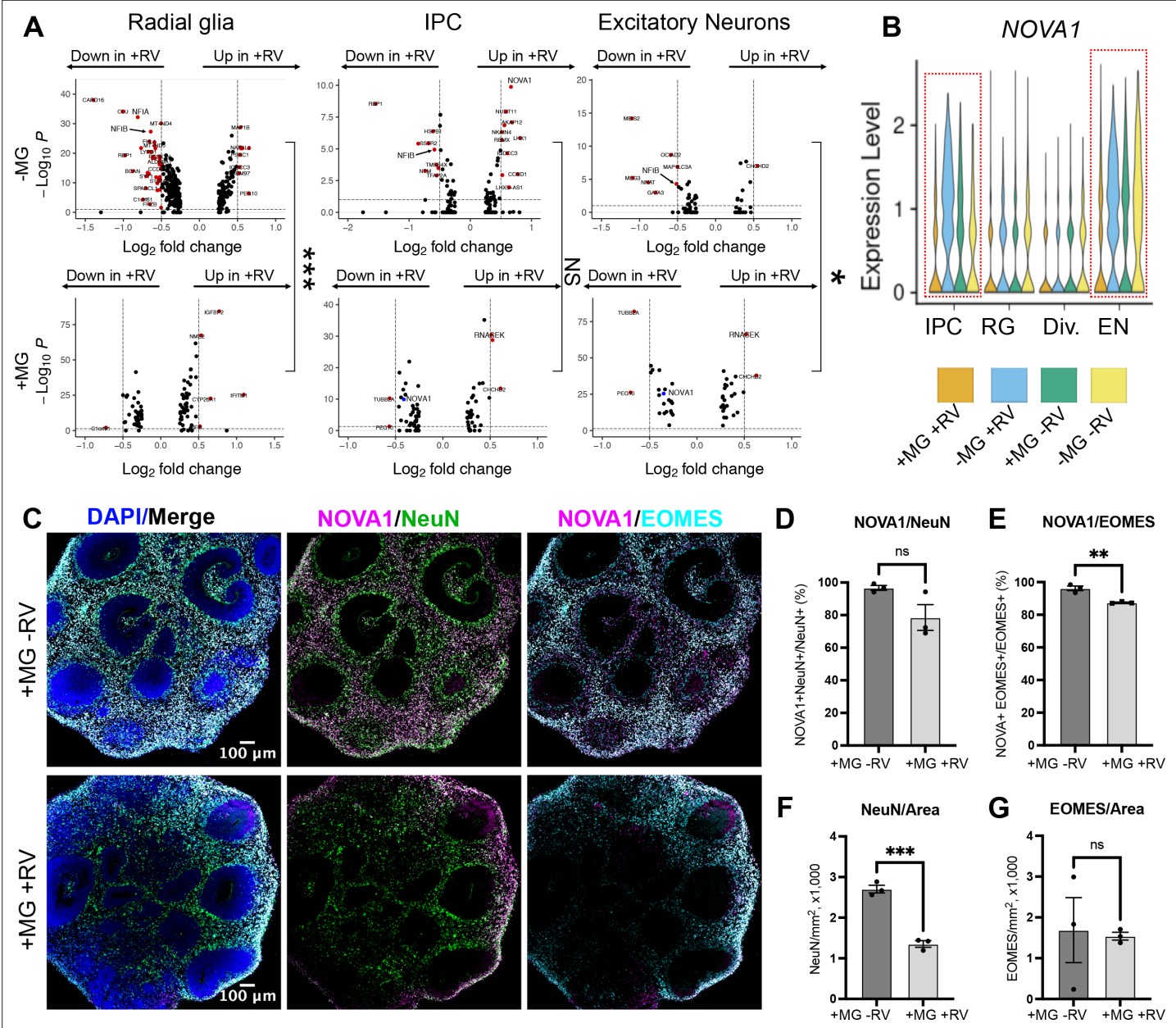

**Figure 6.** NOVA1 expression is reduced in response to rubella virus (RV) exposure. (**A**) Differentially expressed genes in different cell types in response to RV treatment without (top panel) and with microglia (bottom panel). IPCs – intermediate progenitor cells. In the presence of microglia, fewer differentially expressed genes in response to RV treatment were identified across all major cell types. In organoids with microglia, NOVA1 trended toward a decrease in IPCs and neurons (labeled in blue in the panel). Kolmogorov-Smirnov test was used on DEGs with p-value <0.05. ***<0.001, NS – not significant, *<0.05. (**B**) Violin plot for *NOVA1* that is differentially expressed in response to RV and presence of microglia. IPCs – intermediate progenitor cells, RG – radial glia, Div. – dividing cells, EN – excitatory neurons. (**C**) Representative images of RV-exposed organoids with microglia at 2 weeks post-exposure, stained with DAPI for cell nuclei (blue), NOVA1 (magenta), NeuN for neurons (green), and EOMES for intermediate progenitor cells (cyan). (**D**) Cell number quantification for NeuN+ neurons that were also positive for NOVA1 in control (heat-inactivated RV) or RV condition. Averages of 3-5 sections (technical replicates) across 3 organoids (biological replicates, individual data points) where used for quantification in D-G. Unpaired parametric Student's t-test was used to compare the two groups in D–G. p-value = 0.088. (**E**) Cell number quantification for EOMES+ intermediate progenitors that were also positive for NOVA1 in control (heat-inactivated RV) or RV condition. p-value = 0.0042. (**F**) Cell number quantification for NeuN+ neurons per organoid area displayed in 1000 cells × mm² or organoid surface area in control (heat-inactivated RV) or RV condition. p-value = 0.0004. (**G**) Cell number quantification for EOMES+ intermediate progenitor cells per organoid area displayed in 1000 cells × mm² or organoid surface area in control (heat-inactivated RV) or RV condition. p-value = 0.86.

in organoids, where microglia do not develop under standard protocols, the RV-induced interferon response was attenuated in the presence of microglia, suggesting a possible protective role of microglia on other cell types. One limitation of the current work is the lack of information on transcriptional differences in microglia in the context of RV-exposed organoids due to the low number of recovered microglia in scRNAseq experiments. However, our data on molecular changes in neural progenitor cells and neurons, which likely produce the bulk of neurological symptoms seen in CRS, provide a valuable resource for future investigation of congenital viral infections. Our finding that the presence of microglia may reduce RV-associated transcriptional differences across different cell populations may also shed light on neuroimmune consequences of other congenital infections that coincide temporally with phases of microglia population expansion and reduction (*Menassa et al., 2022*).

Interestingly, RV infection rates were largely influenced by the local cell environment, where proximity to non-microglia cells was necessary for RV infection of microglia. This requirement did not depend on cell-to-cell contact or cell type, thus eliminating phagocytosis of infected non-microglia cells or cell type-specific factors as the explanation for enhanced microglia infectivity. It is possible that the non-microglia supporting cells generate a reservoir of virus, though infection of non-microglia cells was limited and it is unclear how these virions would be different from virions in the viral stocks. More likely, diffusible factors contribute to RV infection of microglia, perhaps in conjunction with other ubiquitous cell surface elements. For instance, such factors could alter the activation state of microglia and thereby alter infectivity. Based on previous reports in 2D cell cultures and pathology examination of infected tissues, RV can establish infection in a variety of cell types, suggesting that the viral entry receptor is ubiquitously expressed, or that viral entry is facilitated by cell membrane components and their modifications. Indeed, membrane phospholipids and glycolipids have been shown to participate in viral entry (*Mastromarino et al., 1990*; *Otsuki et al., 2018*). While our study did not directly address the molecular mechanisms of entry, our findings motivate new directions to advance our limited understanding of host factors needed for RV entry and infection. Moreover, our study highlights the importance of considering tissue complexity when studying viral infection in brain organoids. Complex, multi-lineage organoids can now be designed by incorporating vascular or immune cells into differentiation protocols (*Cakir et al., 2022*; *Cakir et al., 2019*; *Popova et al., 2021*). We show that transcriptomic consequences of RV exposure are dependent on the presence of microglia in the organoid tissue environment, while future studies will be needed to determine the precise mechanisms that mediate this effect. One possibility is that microglia become the predominant cellular target of RV infection. Another possibility is microglia actively altering the microenvironment to modulate the antiviral response.

Clearly, efforts to eliminate RV worldwide through vaccination are a priority. However, our work on neuroimmune interactions in CRS may inform how early brain development goes awry in many contexts including prenatal infection with other neurotropic viruses, genetic conditions associated with dysregulated interferon responses such as Aicardi Goutières syndrome, and a variety of perturbations that activate common inflammatory pathways. Understanding the specific role of microglia may be key to unlocking the pathophysiology and developing therapies to prevent or mitigate damage.

## Materials and methods

### Cell lines

Vero cells were obtained from ATCC (CRL-1587) and maintained in DMEM (Thermo Fisher, 11965-118) with 10% (vol/vol) fetal bovine serum (Thermo Fisher, 10438026), 10 mM HEPES (Thermo Fisher, 15630-080), and 1× penicillin/streptomycin (Thermo Fisher, 10378016). Cell cultures were routinely checked to be free from mycoplasma.

### Rubella virus

To generate viral stocks, a plasmid containing a full infectious clone of RV-M33 was linearized then added to an in vitro transcription reaction with Sp6 (New England Biolabs, M0207L). The resulting RNA was purified then polyadenylated (New England Biolabs, M0276S) and capped using Vaccinia Capping System (New England Biolabs, M2080S). This RNA was then introduced to Vero cells using TransIT-mRNA transfection (Mirus Bio, MIR 2250). At 72 hr post-transfection, culture media was collected and passaged onto fresh Vero cells. To generate viral stocks, Vero cells were inoculated

with low passage number RV (P2-P3) and cultured at 37°C. Culture media was collected at 72 hr post-inoculation, clarified, and stored at −80°C. Immunofluorescent titering assays were performed on Vero cells using anti-RV capsid antibody (ab34749), yielding titers of $10^5$–$10^6$ focus-forming units/ml (ffu/ml) for RV stocks. RV-GFP stocks were prepared in the same manner, from a plasmid that had been modified through an in vitro reaction with nCas9 and custom guides to cut the RV-M33 plasmid midway through the p150 gene at residues 717–718 (dgRNA system with DNA oligos for RNA in vitro transcription as follows: tracrRNA sequence: AAA AAG CAC CGA CTC GGT GCC ACT TTT TCA AGT TGA TAA CGG ACT AGC CTT ATT TTA ACT TGC TAT GCT GTC CTA TAG TGA GTC GTA TTA, crRNA oRV012 sequence: CAA AAC AGC ATA GCT CTA AAA CGC TCG CGG CCA CGT CAC CGC CTA TAG TGA GTC GTA TTA). After cutting the plasmid, an sfGFP sequence flanked by Gly-Gly-Ser-Gly-Gly linkers (PCR-amplified using primers oRV010: CTG GCC CCG GCC AGC TCG GAG GAT CGG GCG GAA TGA GCA AGG GCG AGG AG and oRV011: GTG ACG TGG CCG CGA GTC CTC CTG ATC CGC CAG TGA TCC CGG CGG CG) was inserted using InFusion (TakaraBio, 638916). GFP expression of the resulting virus was validated through co-labeling of RV-GFP-infected Vero cells with anti-RV capsid antibody. All viral stocks were tested to be free from mycoplasma.

## Consent statement UCSF

Deidentified tissue samples were collected with previous patient consent in strict observance of the legal and institutional ethical regulations. Protocols related to human iPSCs were approved by the Human Gamete, Embryo, and Stem Cell Research Committee (institutional review board) at the University of California, San Francisco.

## Primary prenatal brain slices

Deidentified primary tissue samples were collected with previous patient consent in strict observance of the legal and institutional ethical regulations. Cortical brain tissue was immediately placed in a sterile conical tube filled with oxygenated artificial cerebrospinal fluid (aCSF) containing 125 mM NaCl, 2.5 mM KCl, 1 mM $MgCl_2$, 1 mM $CaCl_2$, and 1.25 mM $NaH_2PO4$ bubbled with carbogen (95% $O_2$/5% $CO_2$). Blood vessels and meninges were removed from the cortical tissue, and then the tissue block was embedded in 3.5% low-melting-point agarose (Thermo Fisher, BP165-25) and sectioned perpendicular to the ventricle to 300 µm using a Leica VT1200S vibrating blade microtome in a sucrose protective aCSF containing 185 mM sucrose, 2.5 mM KCl, 1 mM $MgCl_2$, 2 mM $CaCl_2$, 1.25 mM $NaH_2PO_4$, 25 mM $NaHCO_3$, 25 mM d-(+)-glucose. Slices were transferred to slice culture inserts (Millicell, PICM03050) on six-well culture plates (Corning) and cultured in prenatal brain slice culture medium containing 66% (vol/vol) Eagle's basal medium, 25% (vol/vol) HBSS, 2% (vol/vol) B27, 1% N2 supplement, 1% penicillin/streptomycin and GlutaMax (Thermo Fisher). Slices were cultured in a 37°C incubator at 5% $CO_2$, 8% $O_2$ at the air-liquid interface created by the cell culture insert.

## Primary human microglia purification

Deidentified primary tissue samples were collected with previous patient consent in strict observance of the legal and institutional ethical regulations. Brain tissue was immediately placed in a sterile conical tube filled with oxygenated artificial spinal fluid containing 125 mM NaCl, 2.5 mM KCl, 1 mM $MgCl_2$, 1 mM $CaCl_2$, and 1.25 mM $NaH_2PO4$ bubbled with carbogen (95% $O_2$/5% $CO_2$). Prenatal human microglia were purified from primary brain tissue from mid-gestation (gestational week 18–23) samples using MACS kit with CD11b magnetic beads (Miltenyi Biotec, 130-049-601) following the manufacturer's instructions. Briefly, primary brain tissue was minced to 1 $mm^3$ pieces and enzymatically digested in 10 ml of 0.25% trypsin reconstituted from 2.5% trypsin (Gibco, 15090046) in DPBS (Gibco, 14190250) for 30 min at 37°C; 0.5 ml of 10 mg/ml of DNAse (Sigma-Aldrich, DN25) was added in the last 5 min of dissociation. After the enzymatic digestion, tissue was mechanically triturated using a 10 ml pipette, filtered through a 40 µm cell strainer (Corning 352340), pelleted at 300×$g$ for 5 min, and washed twice with DBPS. Dissociated cells were resuspended in MACS buffer (DPBS with 1 mM EGTA and 0.5% BSA) with addition of 0.5 mg/ml DNAse and incubated with CD11b antibody for 15 min on ice. After the incubation, cells were washed with 10 ml of MACS buffer and loaded on LS columns (Miltenyi Biotec, 130-042-401) on the magnetic stand. Cells were washed three times with 3 ml of MACS buffer, then the column was removed from the magnetic field and microglia cells were eluted in 5 ml of MACS buffer. Cells were pelleted at 300×$g$, resuspended in 1 ml of culture

media, counted, and used for downstream analysis. We routinely obtained 1×10⁶ of microglia cells from a single MACS purification.

For experiments requiring microglia co-culture with different cell types, the flow-through eluent from microglia selection served either as a cell type fraction depleted of microglia (denoted as 'flow-through') or was used for an additional separation between neuronal and glial fractions by using PSA-NCAM antibody (Miltenyi Biotec, 130-092-966) following the same procedure described for microglia purification.

## 2D microglia cultures

Microglia were cultured on glass-bottom 24-well plates (Cellvis, P24-1.5H-N) pre-coated with 0.1 mg/ml of poly-d-lysine (Sigma-Aldrich, P7280) for 1 hr and 1:200 laminin (Thermo Fisher, 23017015) and 1:1000 fibronectin (Corning, 354008) for 2 hr. Microglia were plated at $1 \times 10^5$ cells/well and maintained in culture media containing 66% (vol/vol) Eagle's basal medium, 25% (vol/vol) HBSS, 2% (vol/vol) B27 (Thermo Fisher, 17504001), 1% N2 supplement (Thermo Fisher, 17502001), 1% penicillin/streptomycin, and GlutaMax (Thermo Fisher) additionally supplemented with 100 ng/ml IL34 (Peprotech, 200-34), 2 ng/ml TGFβ2 (Peprotech,100-35B), and 1× CD lipid concentrate (Thermo Fisher, 11905031) for 5–8 days. For co-culture experiments, other cell types were cultured with microglia at 5:1 ratio ($1 \times 10^5$ microglia cells for each $5 \times 10^5$ non-microglial cells).

## iPSC lines

All work related to human iPSCs has been approved by the UCSF Committee on Human Research and the UCSF GESCR (Gamete, Embryo, and Stem Cell Research) Committee.

Human iPSC line 'WTC-10' derived from healthy 30-year-old Japanese male fibroblasts was from the Conklin Lab, UCSF (*Bershteyn et al., 2017*; *Kreitzer et al., 2013*). Human iPSC line '13325' was derived from 9-year-old female fibroblasts originally obtained from Coriell cell repository.

Human iPSC line '1323-4' derived from healthy 48-year-old Caucasian female fibroblasts (gift from the Conklin Lab, UCSF) was used for immunofluorescence validation analysis as we found that this line generates more reproducible brain organoids.

## Organoid generation

Cerebral organoids were generated based on a previously published method (*Paşca et al., 2015*) with several modifications. Briefly, hiPSCs cultured on Matrigel were dissociated into clumps using 0.5 mM EDTA in Ca²⁺/Mg²⁺-free DPBS and transferred into ultra-low attachment six-well plates in neural induction media (GMEM containing 20% [vol/vol] KSR, 1% [vol/vol] penicillin-streptomycin, 1% [vol/vol] non-essential amino acids, 1% [vol/vol] sodium pyruvate, and 0.1 mM 2-mercaptoethanol). For the first 9 days, neural induction media was supplemented with the SMAD inhibitors SB431542 (5 µM) and dorsomorphin (2 µM), and the Wnt inhibitor IWR1-endo (3 µM). Additionally, the Rho kinase inhibitor Y-27632 (20 µM) was added during the first 4 days of neural induction to promote survival. Neural induction media was replaced every 2 days for 8 days, and Y-27632 was removed from the media on the fourth day. After neural induction, plates containing cortical organoids were transferred to a plate shaker rotating at 80 rpm. Between days 9 and 25, organoids were transferred to an expansion media (1:1 mixture of Neurobasal and DMEM/F12 containing 2% [vol/vol] B27 without vitamin A, 1% N2, 1% [vol/vol] non-essential amino acids, 1% [vol/vol] GlutaMax, 1% [vol/vol] antibiotic/antimycotic, 0.1 mM 2-mercaptoethanol) supplemented with FGFβ (10 ng/ml) and EGF (10 ng/ml). Between days 25 and 35, organoids were maintained in neural differentiation media without FGF or EGF. From day 35 onward, organoids were maintained in neural differentiation media containing B27 with vitamin A with full media exchanges every 2–3 days.

## Microglia-organoid engraftment and co-culture

Microglia from mid-gestation cortical tissue were MACS-purified and immediately added to organoids between weeks 5 and 6 in six-well plates at $1 \times 10^5$ microglia cells/organoid and kept off the shaker overnight. The following day, the plates were returned to the shaker and maintained following a usual organoid maintenance protocol.

## RV infection

Cells cultured in 2D were inoculated by adding RV stock virus to culture media in 1:1 dilution (250 µl of media to the equal volume of viral stock, $1.75 \times 10^5$ total ffu/well) to achieve a multiplicity of infection

(MOI) of 2. After 4 hr, media was exchanged with fresh cell culture media. Cortical brain slices were treated with 500 µl of RV viral stock ($3.5 \times 10^5$ total ffu/slice) applied over the slice culture filter for 4 hr, and then the viral culture media was removed and replaced with fresh slice culture media. Organoids were treated in six-well plates with 2 ml of 1:1 dilution of viral stock:organoid maintenance media ($7 \times 10^5$ total ffu) for 4 hr, and then viral media was exchanged for fresh media. For all experimental conditions, cells were fixed and processed for downstream analysis at 72 hr post-infection. Supernatant from non-infected Vero cells (mock) or heat-inactivated RV (65°C, 30 min) was used as control.

For titering experiments, microglia co-cultures or Vero cells (as controls) were infected at the indicated MOI. Cells were inoculated for 4 hr, then fresh media was replaced, and sampled at the indicated timepoints. Media samples were clarified and flash frozen. Viral titer was then quantified in the media samples using immunofluorescence titering assay.

## Immunofluorescence

Cells cultured on glass-bottom well plates were fixed in 4% PFA at the room temperature for 10 min and washed with PBS three times for 5 min each wash. Blocking and permeabilization were performed in a blocking solution consisting of 10% normal donkey serum, 1% Triton X-100, and 0.2% gelatin for 1 hr. Primary and secondary antibodies were diluted and incubated in the blocking solution. Cell cultures were incubated with primary antibodies at the room temperature for 1 hr, washed 3× with washing buffer (0.1% Triton X-100 in PBS), and incubated with secondary antibodies for 1 hr at the room temperature.

Organoid samples were fixed in 4% PFA at the room temperature for 1 hr. Whole organoids were incubated in 30% sucrose (wt/vol) at 4°C overnight, cryopreserved in OCT/30% sucrose (1:1), and then cryosectioned at 20 µm thickness. Blocking and permeabilization were performed in a blocking solution consisting of 10% normal donkey serum, 1% Triton X-100, and 0.2% gelatin for 1 hr. Primary and secondary antibodies were diluted and incubated in the blocking solution. Cryosections were incubated with primary antibodies at 4°C overnight, washed 3× for 10 min each with washing buffer (0.1% Triton X-100 in PBS). Slides were incubated with species-specific Alexa Fluor secondary antibodies (1:2000) overnight at 4°C and then washed with washing buffer for at least 3× for 10 min each. Finally, slices were mounted with glass coverslips using DAPI Fluoromount-G (Southern Biotech, 0100-20) mounting media.

Cortical slices were fixed in 4% PFA at room temperature for 1 hr. Antibody staining was performed as for organoid samples above, with the exceptions that no cryosectioning was performed.

Images were collected using Leica SP8 confocal system with 20× air lens (0.75 NA) and 63× oil lens (1.40 NA). Images were processed using ImageJ/Fiji and Affinity Designer software.

## Antibodies

Primary antibodies used in this study included: rabbit Iba1 (1:500, Wako, 019-19741), guinea pig Iba1 (1:500, Synaptic Systems, 234 004), mouse RV capsid (1:500, Abcam, ab34749), rat Sox2 (1:500, Invitrogen, 14-9811-82), chicken GFP (1:1000, Aves labs, GFP-1020), mouse NOVA1 (1:500, Santa Cruz, sc100334), rabbit EOMES (1:200, Sigma-Aldrich, HPA028896), chicken NeuN (1:1,000, Millipore, ABN91), rabbit IFITM3 (1:500, Proteintech, 11714-1-AP).

## Organoid single-cell capture for scRNAseq

Two organoids per experimental condition were washed with $Ca^{2+}/Mg^{2+}$-free DPBS and cut into 1 mm$^2$ pieces and enzymatically digested with papain digestion kit (Worthington, LK003163) with the addition of DNAse for 1 hr at 37°C. Following enzymatic digestion, organoids were mechanically triturated using a P1000 pipette, filtered through a 40 µm cell strainer test tube (Corning 352235), pelleted at 300×*g* for 5 min, washed twice with DBPS, and resuspended in 180 µl of DPBS on ice for barcoding with MULTI-seq indices (*McGinnis et al., 2019*) for multiplexing. Anchor and barcoded strands unique for each sample were mixed in 1:1 molar ratio in DPBS (without BSA or FBS to avoid sequestering labeling oligonucleotides) and 20 µl of 10× Anchor:Barcode mixture was added to 180 µl of cell suspension. Cells were incubated on ice for 5 min, and then 20 µl of co-anchor was added to each tube. Cells were incubated on ice for additional 5 min and washed with ice-cold 1% BSA in DPBS. Cells were counted and kept on ice to prevent barcode loss. Two organoid lines with and without microglia that were treated with RV or uninfected Vero cell supernatant were combined and captured across

seven lanes of 10x Genomics using Chromium Single Cell 3' Reagent Kit (v2 Chemistry) following the manufacturer's protocol.

scRNAseq libraries were generated using the 10x Genomics Chromium 3' Gene Expression Kit. Briefly, barcoded single-cell mixtures from different conditions ranging from two to three individual conditions per lane were loaded onto chromium chips with a capture target of 10,000 cells per sample. The 10× protocol was modified for collection of MULTI-seq barcodes. During SPRI cleanup immediately following cDNA amplification, supernatant was saved to recover the barcode fraction. Endogenous transcript cDNA remained bound to the SPRI beads and the protocol was continued for endogenous transcripts without change. Libraries were prepared following the manufacturer's protocol and sequenced on an Illumina NovaSeq with a targeted sequencing depth of 50,000 reads per cell. BCL files from sequencing were then used as inputs to the 10x Genomics Cell Ranger pipeline.

## MULTI-seq barcode amplification

Supernatant collected after cDNA amplification cleanup step was transferred to fresh 1.5 ml Eppendorf tubes, and 260 µl SPRI (for a final ratio of 3.2×) and 180 µl 100% isopropanol (for a final ratio of 1.8×) were added. After pipette mixing 10 times, the solution was incubated at room temperature for 5 min, placed on magnetic rack for solution to clear. The supernatant was removed, and the beads were washed with 500 µl of 80% ethanol twice. Air-dry beads were removed from magnet, resuspended in 50 µl buffer EB. After clearing the solution on the magnet, supernatant was transferred to a new PCR strip. Libraries were prepared with KAPA HiFi master mix with universal I5 primers and RPI primers unique for each 10× lane. PCR was performed with the following protocol: 95°C for 5 min (98°C for 15 s, 60°C for 30 s, 72°C for 30 s) repeated for 10 times, 72°C for 1 min, 4°C hold.

PCR product was cleaned with 1.6× SPRI beads and resuspended in 25 µl buffer EB. Barcode libraries were quantified at 1:5 concentration using Bioanalyzer High Sensitivity DNA analysis. Barcodes were sequenced as fraction of endogenous cDNA library with a target of 3000–5000 barcode reads per cell.

## scRNAseq analysis

CellRanger 3.0 was used to create a cell by gene matrix which was then processed using Solo (*Fleming et al., 2019*) for doublet detection and removal. A minimum of 1000 genes, 500 UMI counts, and 20% mitochondrial cutoff were used to remove low-quality cells from all datasets. MAST (*Finak et al., 2015*) was used on log normalized raw counts for all differential expression tests. The gene marker lists were filtered after testing by specifically removing unannotated genes from HGNC. Organoid demultiplexing and doublet filtering was done through deMULTIplex (*McGinnis et al., 2019*) (https://github.com/chris-mcginnis-ucsf/MULTI-seq ; *McGinnis, 2019*). Uniform manifold approximation and projection (UMAP) (*McInnes et al., 2018*) embeddings and neighbors for Leiden clustering (*Traag et al., 2019*) were used for clustering and visualization. Nebulosa was used to generate density plots and (*Bunis et al., 2020*) for color-blind friendly plotting of clusters. Pearson correlation was calculated on the intersection of the shared genes between datasets which averaged Pearson residuals for each cluster. Organoid cells were batch corrected using default parameters of the SCTransform (*Hafemeister and Satija, 2019*) integration workflow.

## Image analysis and statistical tests

### 2D microglia co-cultures

Cell co-localization with the RV capsid was quantified using the CellProfiler 3.0 software (*McQuin et al., 2018*). First, individual cells were identified by using IdentifyPrimaryObjects module with threshold strategy 'Global', threshold method 'Otsu', and a two-class thresholding for each individual channel for DAPI, Iba1 and RV capsid fluorescence images. Then, resulting cell objects were paired by using RelateObjects module to identify Iba1-postive, RV-positive and double-positive DAPI objects. Finally, CalculateMath was used to quantify proportions for each cell population, including RV-positive and RV-negative Iba1 microglia cells and non-microglia cells, depending on the analysis.

### Organoid immunofluorescence quantifications

For cytoplasmic IFITM3 staining, whole organoid sections were treated as regions of interest and average fluorescence intensity for IFITM3 was normalized to DAPI fluorescence using QuPath 0.3.2. To conduct IFITM3 intensity quantifications, the entire organoid section was defined as region of

interest (ROI) by using the wand tool. Intensity calculation features with the analyze function for the DAPI and IFITM3 channels were then used to determine ROI fluorescence intensities. To set up the calculation, preferred pixel size was set to 0.76 µm according to image resolution. Relevant measurements including intensity mean, standard deviation, min and max, and organoid area were calculated and retrieved from QuPath detection measurements.

For analyzing nuclear signal for NOVA1/EOMES/NeuN experiments, QuPath 0.3.2 was used to identify and quantify individual cell nuclei. First, each organoid was selected as ROI by using the wand tool. To establish total EOMES count in each organoid, the cell detection function was applied to the appropriate channel where adjustments were made, including thresholding and deselecting. The positive cell detection function with the EOMES channel thresholding value were then applied on NOVA1 channel to identify cells that are NOVA1+/EOMES+ double positive. Thresholding adjustments were made to account for imaging variations in EOMES channel. To quantify NOVA1+/NeuN+ nuclei, the same two-step detection procedure described above was utilized with NOVA1 and NeuN channels. All relevant quantification values were then collected from QuPath detection measurements. Three to five organoid sections per each organoid were imaged and analyzed, with each data point representing an average of several sections per individual organoid.

Prism 9.3.1 was used for statistical analysis and data plotting. Unpaired t-test with assumed Gaussian distribution of the variants and the same standard deviations were used to calculate statistical significance for cell counts. Unpaired nonparametric Kolmogorov-Smirnov test was used to compare differentially expressed genes that reached significance value of $p=0.05$ between conditions in organoids.

Parts of figure schematics were done using https://www.biorender.com/.

## Acknowledgements

We thank Tom Hobman for generously sharing reagents for the Rubella M33 strain, and all members of the Nowakowski and DeRisi laboratories for helpful discussions and comments throughout this project. We would like to thank UCSC Cell Browser and especially Maximilian Haeussler and Brittney Wick for making the single-cell data publicly available. This study was supported in part by gifts from Schmidt Futures and the William K Bowes Jr. Foundation, Simons Foundation grant (SFARI 491371 to TJN), Chan Zuckerberg Biohub Intercampus Investigator Award, NARSAD Young Investigator Grant (to TJN), NINDS award R01NS123263 (to TJN), and NRSA F32 1F32MH118785 (to GP), NINDS F31NS108615 (to HR), UCSF Discovery Fellows Program (to HR), and the Chan Zuckerberg Biohub (to JD). TJN is a New York Stem Cell Foundation Robertson Neuroscience Investigator.

## Additional information

### Funding

| Funder | Grant reference number | Author |
| --- | --- | --- |
| National Institutes of Health | R01NS123263 | Tomasz Nowakowski |
| National Institutes of Health | 1F32MH118785 | Galina Popova |
| National Institutes of Health | 5F31NS108615 | Hanna Retallack |
| Simons Foundation | SFARI 491371 | Tomasz Nowakowski |
| Chan Zuckerberg Biohub | Intercampus Investigator Award | Tomasz Nowakowski Joseph L DeRisi |
| NARSAD | Young Investigator Grant | Tomasz Nowakowski |
| Schmidt Futures | | Tomasz Nowakowski |
| William K. Bowes, Jr. Foundation | | Tomasz Nowakowski |

| Funder | Grant reference number | Author |
|---|---|---|
| University of California, San Francisco | Discovery Fellows Program | Hanna Retallack |

The funders had no role in study design, data collection and interpretation, or the decision to submit the work for publication.

## Author contributions

Galina Popova, Conceptualization, Formal analysis, Investigation, Methodology, Writing - original draft, Writing – review and editing; Hanna Retallack, Conceptualization, Resources, Formal analysis, Investigation, Visualization, Methodology, Writing - original draft, Writing – review and editing; Chang N Kim, Data curation, Formal analysis; Albert Wang, Investigation; David Shin, Resources, Investigation; Joseph L DeRisi, Conceptualization, Resources, Supervision, Methodology, Writing – review and editing; Tomasz Nowakowski, Conceptualization, Supervision, Funding acquisition, Writing – review and editing

## Author ORCIDs

Galina Popova (ID) http://orcid.org/0000-0001-8249-219X
Hanna Retallack (ID) http://orcid.org/0000-0003-0533-9102
Albert Wang (ID) http://orcid.org/0000-0001-5989-4617
Tomasz Nowakowski (ID) http://orcid.org/0000-0003-2345-4964

## Ethics

De-identified tissue samples were collected with previous patient consent in strict observance of the legal and institutional ethical regulations. Protocols related to human iPSC cells were approved by the Human Gamete, Embryo, and Stem Cell Research Committee (institutional review board) at the University of California, San Francisco.

Reviewer #1 (Public Review): https://doi.org/10.7554/eLife.87696.3.sa1
Reviewer #2 (Public Review): https://doi.org/10.7554/eLife.87696.3.sa2
Author Response https://doi.org/10.7554/eLife.87696.3.sa3

# Additional files

## Supplementary files

• MDAR checklist

## Data availability

Sequences of RV and RV-GFP have been deposited at Genbank under accessions OM816674 and OM816675 respectively. Single-cell RNA-seq data for iPSC-derived organoids are available from Gene Expression Omnibus (GEO) under the accession code GSE232462. Processed single-cell data, including dimensionality reduction object, is freely available at https://cells.ucsc.edu/?ds=rubella-organoids. Code associated with analysis of the single cell analysis can be accessed at Github: https://github.com/cnk113/analysis-scripts (copy archived at *Kim, 2023*).

The following datasets were generated:

| Author(s) | Year | Dataset title | Dataset URL | Database and Identifier |
|---|---|---|---|---|
| Retallack H, DeRisi J | 2022 | Synthetic construct clone M33, complete sequence | https://www.ncbi.nlm.nih.gov/nuccore/OM816674 | NCBI GenBank, OM816674 |
| Popova G, Kim CN, Nowakowski TJ | 2023 | Rubella virus tropism and single cell responses in human primary tissue and microglia-containing organoids | https://www.ncbi.nlm.nih.gov/geo/query/acc.cgi?acc=GSE232462 | NCBI Gene Expression Omnibus, GSE232462 |

*Continued on next page*

*Continued*

| Author(s) | Year | Dataset title | Dataset URL | Database and Identifier |
|---|---|---|---|---|
| Retallack H, DeRisi J | 2023 | Synthetic construct clone M33-p150GFP, complete sequence | https://www.ncbi.nlm.nih.gov/nuccore/OM816675 | NCBI GenBank, OM816675 |

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
