## [Editor Report · eLife assessment]

The manuscript represents an **important** study on the pathogenesis of rubella virus tropism and neuropathology in human microglia-containing human stem cell derived organoids and human fetal brain slices. The strength of evidence is **compelling**, employing two different human-relevant models. The findings will be of broad interest to virologists and infectious disease experts, as well as neurodevelopmental biologists. The findings could also be of interest to pediatrics and obstetrics clinical colleagues.

---

## [Referee Report · Reviewer #1 (Public Review)]

The authors sought to address the longstanding question of which cell types are infected during congenital or perinatal rubella virus infection. They used brain slice and organoid-microglia experimental models to demonstrate that the main cell types targeted by rubella virus are microglia. The authors further show that infection results in augmented interferon responses in neighboring neuronal cells but not in the microglia themselves. The data convincingly support the conclusions, with major strengths being the sophisticated primary cell models and single-cell RNA-Seq used to pinpoint microglia as the main cellular targets of rubella virus, and neurons as the bystander targets of immune signaling. This study reveals a new cellular target that will have important implications for basic studies on rubella virus-host interactions and for the potential development of therapies or improved vaccines targeting this virus. As rubella virus is a pathogen of high concern during human pregnancy, this study is also relevant in the field of neonatal infectious diseases.

---

## [Referee Report · Reviewer #2 (Public Review)]

Maternal infection by Rubella virus (RV) early during pregnancy is a serious threat to the health of the fetus. It can cause brain malformation and later expose the newborn to a constellation of symptoms collectively named Congenital Rubella Syndrome (CRS). In this manuscript, the authors provide novel exciting findings on the pathophysiological mechanisms of RV infection during human brain development. By combining analyses of human fetal brain material and cerebral organoids, Popova and colleagues uncovered a selective tropism of RV for microglial cells. Their results suggest that the infection of microglia by RV relies on the presence of diffusible factors secreted by neighboring brain cells. Moreover, the authors showed that RV infection of human cerebral organoids leads to a strong interferon response and dysregulation of neurodevelopmental genes, which both may contribute to brain malformation. Altogether, these data shed some new light on the cellular tropism and the pathophysiological mechanisms triggered by RV infection in the developing brain. This study also raises the importance of using human cell-based models to further understand the pathophysiological mechanisms of CRS. Identifying the cellular and molecular targets of Rubella virus will also pave the way to develop therapies against CRS.

---

## [Author Response]

**Author Response**

The following is the authors’ response to the original reviews.

We thank the reviewers for their time in evaluating the strengths and weaknesses of our manuscript.

We are pleased to see that all reviewers recognized the high significance of our work, noting that the manuscript addresses “longstanding question of which cell types are infected during congenital or perinatal rubella virus infection”. As noted by reviewer 1, “This study reveals a new cellular target that will have important implications for basic studies on rubella virus-host interactions and for the potential development of therapies or improved vaccines targeting this virus. As the rubella virus is a pathogen of high concern during human pregnancy, this study also has important implications in the field of neonatal infectious diseases”.

Below, we provide responses (in blue) to specific critiques:

**Reviewer #1 (Public Review):**

A weakness is that the current data do not provide information on the full replicative potential of the rubella virus in microglia, or whether the virus persists in this system.

See our response below. Briefly, we include new experimental evidence from primary tissue, microglia-transplanted organoids, and Vero cells to further characterize the dynamics of viral infection.

**Reviewer #1 (Recommendations for the authors):**

Most of the viral assays in the brain slices and organoids examine viral protein synthesis, which is a surrogate for genome replication. However, basic virological characterization is lacking and would improve the robustness of the model and its potential utility to understand better rubella virus-microglia interactions. Questions the authors should consider with new experiments include:

Are new virions produced? Can viruses be detected in the media?

Or, are the infections abortive, with viral protein synthesis occurring, but no virus production?

We performed RV titering experiments in dissociated microglia co-cultured with other cell types, as well as Vero cells as a control. While we can detect a robust increase in viral titer from Vero cells, it fell below detection levels in microglia co-cultures. See Author response image 1. We now include these data in Supplementary Figure 2D.

**Author response image 1. sa3fig1:** Rubella virus titering experiment performed in Vero cells (positive control) or dissociated microglia co-cultures. In primary microglia co- cultures, viral titer falls below detection levels after several days of infection.

While we could not detect an increase in the viral particles from microglia mixed cultures, we confirmed the presence of GFP from the RV-GFP reporter construct, and we believe it serves as a proof that the virus can infect microglia cells and lead to production of functional viral protein (Author response image 2, Figure 1E-F of the current manuscript):

**Author response image 2. sa3fig2:** 

We also observed an increase in RV RNA over time in tissue slice infections, using qPCR (Author response image 3, not included in the manuscript).

**Author response image 3. sa3fig3:** Modest increase in RV RNA over time in brain slice infections. Rubella virus RNA measured by qPCR relative to GAPDH gene, in n=3 samples (2 technical replicates each condition). Brain slices were exposed to RV, then collected at end of inoculation (4 hours post infection), or at 3 or 5 days post infection, and processed for RNA extraction and RT-qPCR.

How long do the infections persist in the model? What is the fate of infected microglia over time? Time courses to monitor infection and cell health would be useful.

We performed a longer infection with RV in organoids transplanted with microglia, and after two weeks of infection, we can detect multiple microglia cells positive for the RV capsid. These data are now included in Figure 4 of the current manuscript.

**Author response image 4. sa3fig4:** After 2 weeks post infection, microglia remain positive for RV capsid.

**Reviewer #2 (Public Review):**

Weaknesses

The set of data is rather descriptive. It suggests that microglia are the predominant brain target of RV in vivo, without identifying the targeting mechanism that provides cell type specificity. Moreover, what are the diffusible cues released from the brain environment that increase microglia infection and RV replication?

We agree with the reviewer that identifying molecular mechanisms that underlie this phenotype will be very interesting to explore in future research, and we acknowledge the limitation of the study in the Discussion.

It is unclear why brain organoids not supplemented by microglia are susceptible to RV inoculation.

We could not detect RV capsid in organoids without microglia after 72 hours of inoculation. We attribute any changes seen at the level of single cell transcriptomics in the absence of microglia transplantation to exposure to virus-associated particles, including but not limited to viral RNA species, viral proteins, or even other components of the viral stocks made in Vero cells. These factors may induce transcriptomic differences even in the absence of RV infection. In the text, we take care to refer to these condition as “Rubella virus-exposed” rather than “Rubella virus- infected”. We now include the following panel from Author response image 5 in Figure 4B of the current manuscript.

**Author response image 5. sa3fig5:** After 2 weeks post infection, microglia remain positive for RVOrganoids without microglia do not show positive RV immunofluorescence.

**Reviewer #2 (Recommendations for the authors):**

Several points could be further addressed to improve the data set and shed more light on some aspects of this manuscript:

• Figure 1. Additional microglia markers should be used to reinforce the evidence that microglia cells are the principal RV targets. Since Iba1 is a marker of activated microglia, does RV have a selective tropism to all microglia or only to activated ones in human fetal brain slices?

The reviewer brings up an interesting point that, in our mind, can be separated into two independent questions:

Are Iba1-positive cells bona fide microglia, or are there other cell populations of macrophage/monocyte origin that are labeled with Iba1? Therefore, additional markers should be used for immunolabeling;Is RV infection selective for microglia “activation” status, when only 5mmune-primed cells can be infected?

For the first point, we have previously shown that in the developing human brain, virtually all Iba1-positive cells are also P2RY12-positive (unpublished; Author response image 6). Therefore, in primary human brain slices, there is a negligible amount of non-microglia macrophages. However, in culture microglia quickly lose their “homeostatic” identity, including P2RY12 expression, as quickly as six hours after ex vivo extraction (Gosselin et al., 2017; DOI: 10.1126/science.aal3222).

**Author response image 6. sa3fig6:** P2RY12 co-localizes with Iba1 in primary brain tissue from gestational week 17.5, including cells with more ameboid morphology (arrows).

However, in organoids at 2 weeks post-RV exposure, we found microglia with both ameboid and more ramified morphology (Author response image 7). It would be challenging and beyond the scope of this manuscript to use morphology or Iba1 intensity levels to determine cause and effect as microglia activation state relates to RV infectivity (i.e. do activated microglia preferentially get infected with the virus, or do infected microglia become activated and upregulate Iba1 levels and change morphology).

**Author response image 7. sa3fig7:** Examples of microglia with round (top) and ramified (bottom) morphology that co-localize with RV capsid staining.

Regarding RV tropism in the 2D culture of microglia, some Iba- cells are infected by RV as they show capsid staining. What are these cells? Are neurons and/or glia also susceptible to RV in vitro infection? Are non-microglial cells getting RV infected in the absence of microglia?

In the absence of microglia cells, a small proportion of non-microglia cells get infected with RV. There is no statistically significant difference in the number of cells that get infected with RV in the presence or absence of microglia across different cell types. We add these data as Supplement Figure 3.

**Author response image 8. sa3fig8:** Rubella infection in non-microglia cells. (**A**) Representative images of different cell types depleted of microglia. Cell cultures were stained RV capsid (green) and DAPI. (**B**) Quantification of total cells that are positive for RV capsid across conditions. (**C**) Quantification of RV+ cells that are not microglia across different cell populations. No statistically significant difference was detected in RV infectivity in cells c-cultured with or without microglia.

• Figure 3. The low rate of Rubella virus infection in homogenous CD11b+ cell culture raises the question of whether the Rubella virus can infect microglia at a specific activation stage. It is also surprising that there is no infection of such cell population (also CD11b+) alone while cultured in 2D, as reported in figure 2. Why such a difference?

It is well established that culture of microglial cells isolated from brain tissue alters their molecular properties, which likely alters the cell surface protein composition. In the revised discussion, we include activation as a possible mechanism that will require further investigation.

• Fig 4A-B, it is unclear whether organoids that are not engrafted with microglia get infected upon RV (with active viral replication) inoculation. If non-microglia-supplemented organoids are indeed infected and allow RV replication, this suggests that organoids might not be the ideal system to model human fetal brain RV infection at GW18-23.

We could not detect RV capsid in organoids without microglia after 72 hours of inoculation. We include the following panel from Author response image 9 in Figure 4 now.

**Author response image 9. sa3fig9:** Organoids without microglia do not show positive RV immunofluorescence.

• Figure 4E, why are cells derived from microglia-free organoids so much enriched in the UMAP plots as compared to the other organoid condition? Is RV impacting cell fitness, proliferation, or neurodifferentiation?

This perceived difference is due to data presentation. Based on cell proportions, cells from organoids that were treated with microglia are more represented in the scRNAseq data, and this difference most likely comes from user-introduced imbalance in cell loading and possible cell losses during demultiplexing (Author response image 10, panel A). Cell number composition across different conditions and cell types, including RV and MG treatment, are shown in Supplement Figure 4 of the current manuscript (Author response image 10, panel B).

**Author response image 10. sa3fig10:** Data composition depending on condition. A. Cell number contribution from organoids with and without microglia. B. Contribution of each condition to each cluster composition.

Contribution of each condition can be visualized via UCSC single cell data browser: https://cells.ucsc.edu/?ds=rubella-organoids

• Figure 4F-H. If microglia is the predominant target for RV in the brain, why are microglia-free organoids susceptible to RV and who are the other cellular targets, whose infection leads to activation of interleukin pathway genes and dysregulation of brain developmental markers in selected subpopulations (RGCs, ENs..).

Thank you for bringing this point. We did not detect any appreciable RV genomic RNA in our published single cell data, nor did we identify RV capsid in the RV-exposed organoids without microglia. Our experiments on dissociated cell cultures show that a small population (~1-4%) of other cell types was positive for the RV capsid, including neuron-enriched and glial-enriched fractions (Author response image 11; Supplementary Figure 3C in current manuscript). We expect a similar proportion of non-microglia cells to be infected in the brain organoids. One possible explanation for the robust interferon response even in the absence of productive infection in other cell types is exposure to virions and virus-associated particles, including but not limited to viral RNA species, viral proteins, or even other components of the viral stocks made in Vero cells (which is a cell line that should not produce interferons, but may produce other unmeasured cytokines as a virally infected cell culture).

**Author response image 11. sa3fig11:** Quantification of RV+ cells that are not microglia across different cell populations. No statistically significant difference was detected in RV infectivity in cells cultured with or without microglia.

• QRT-PCR validations of some of these key brain targets should be performed.

We agree with the reviewer that further validation of the predicted molecular changes downstream of Rubella exposure would be valuable. We have opted to validate IFITM3 and NOVA1 expression differences using immunostaining, and the results are consistent with our predictions from scRNAseq, and the data is presented in revised Figure 5 and 6 of the current manuscript.

**Reviewer #3 (Public Review):**

Weaknesses of the paper: Overall, additional control experiments are needed to support the stated conclusions. Affinity chromatography is used to purify microglia and other cell types, but the overall cell enrichment is not quantified.

We appreciate the reviewer concern. However, affinity based enrichments rarely guarantee purity of the enrichment, and we do not believe accurate estimation of the purification purity would alter the biological interpretation of the data.

In cell mixing experiments, the authors do not rule out the possibility that the added non- microglia cells also become infected, releasing additional infectious viruses. The finding that a diffusible factor is required for RV infection would be unusual if not unprecedented; therefore, additional data are required to support this claim and rule out other interpretations.

We provide quantification of non-microglia cells that are positive for RV capsid in the presence and absence of microglia. Small (~1-4%) of non-microglia cells get infected with the virus and can potentially release more of the virus (see Author response image 12), but we do not know how this newly produced virus would be different from the one that was applied to the cells directly. To follow up our co-culture experiments, we wanted to exclude a possibility of microglia engulfing RV- infected cells in co-cultures, therefore we separated the two cell fractions by a liquid-permeable membrane (Figure 3 of the current manuscript). It is possible that factors secreted by other cell populations in the transwell assay experiments act on microglia cells to upregulate a yet unidentified receptor on microglia surface or other infection-dependent molecule rendering them infectable by the virus.

**Author response image 12. sa3fig12:** Rubella infection in non-microglia cells. (**A**) Representative images of different cell types depleted of microglia. Cell cultures were stained RV capsid (green) and DAPI. (**B**) Quantification of total cells that are positive for RV capsid across conditions. (**C**) Quantification of RV+ cells that are not microglia across different cell populations. No statistically significant difference was detected in RV infectivity in cells c-cultured with or without microglia.

We re-phrase the text by de-emphasizing “soluble factors” and focusing on excluding phagocytosis of infected cells as a possible mechanism of RV capsid immunoreactivity in microglia cells.

The methods section would be improved by including details about the iPSC line that was used.

We include the following section in Materials and Methods:

iPSC lines.

All work related to human iPS cells has been approved by the UCSF Committee on Human Research and the UCSF GESCR (Gamete, Embryo, and Stem Cell Research) Committee. Human iPS cell line “WTC-10” derived from healthy 30-year-old Japanese male fibroblasts was from the Conklin Lab, UCSF (Bershteyn et al., 2017; Kreitzer et al., 2013). Human iPSC line “13325” was derived from 9-year-old female fibroblasts originally obtained from Coriell cell repository. Human iPSC line “1323-4” derived from healthy 48-year-old Caucasian female fibroblasts (gift from the Conklin Lab, UCSF) was used for immunofluorescence validation analysis as we found that this line generates more reproducible brain organoid differentiations.

and by a more thorough description of virus-specific details, including the numbers of infectious particles added per volume of incubation media.

We now include the following data in the Materials and Methods section:

Rubella virus infection

Cells cultured in 2D were inoculated by adding RV stock virus to culture media in 1:1 dilution (250 ul of media to the equal volume of viral stock, 1.75x105 total ffu/well) to achieve a multiplicity of infection (MOI) of 2. After four hours, media was exchanged with fresh cell culture media. Cortical brain slices were treated with 500 ul of RV viral stock (3.5x105 total ffu/slice) applied over the slice culture filter for four hours, and then the viral culture media was removed and replaced with fresh slice culture media. Organoids were treated in 6-well plates with 2ml of 1:1 dilution of viral stock:organoid maintenance media (7x105 total ffu) for four hours, and then viral media was exchanged for fresh media. For all experimental conditions, cells were fixed and processed for downstream analysis at 72 hours post infection. Supernatant from non-infected Vero cells (mock) or heat-inactivated RV (650C, 30 mins) was used as control.

In addition to immunofluorescence, adding additional data to demonstrate and quantify virus infection (PCR and plaque assays. or immunofluorescence using an anti-double-stranded RNA antibody such as J2) from the infected brain slices and organoids would provide greater assurance that the virus is indeed replicating under the experimental conditions.

We performed RV titering experiment in dissociated microglia co-cultured with other cell types, as well as Vero cells control. While we can detect a robust increase in viral titer from Vero cells, it fell below detection levels in microglia co-cultures. We now include these data in Supplementary Figure 2D.

**Author response image 13. sa3fig13:** Rubella virus titering experiment performed in Vero cells (positive control) or dissociated microglia co-cultures. In primary microglia co- cultures, viral titer falls below detection levels after several days of infection.

Unfortunately, we did not find J2 staining informative because we could detect signal in both wild type RV infection conditions and in heat-inactivated RV, presumably due to native dsRNA species present in cells. We did not detect any increase or difference in the pattern of staining between RV and heat-inactivated virus-exposed conditions (Author response image 14; not included in the manuscript).

**Author response image 14. sa3fig14:** J2 antibody labels dsRNA in both RV-exposed and control heat- inactivated virus conditions, presumably due to native dsRNA that is not unique to the viral replication.

Organoid imaging with immunofluorescence would be very informative in demonstrating the presence of microglia and also in showing which cells are virus-infected in the context of organoid structures.

We provide images from 72hrs and 2 week RV infection, providing a zoomed-out view of organoids with microglia and RV capsid staining. We also provide images of 72hrs post- infection in organoids without microglia Author response image 15 (Figure 4C in current manuscript).

**Author response image 15. sa3fig15:** Microglia in organoids co-localize with RV capsid staining.

GenBank accession numbers are listed for the recombinant RV and GFP-RV reporter, but a search using those numbers did not locate the deposits--perhaps the deposits were very recent?

Both viral construct information is now available on GenBank:

M33 RV strain can be found here: https://www.ncbi.nlm.nih.gov/nuccore/OM816674

RV-GFP can be found here: https://www.ncbi.nlm.nih.gov/nuccore/OM816675

The authors incorrectly refer to the GFP virus as a new strain; it is not a viral strain and should be referred to as a reporter virus.

Thank you, we changed the description to

“To confirm functional transcription and translation of the viral genome, a new reporter construct of RV designed to express GFP within the non-structural P150 gene was generated (RV-GFP, GenBank Accession OM816675)”

Given that the authors show that Vero cell cultures are infected by the Rubella virus in the absence of other cells, additional evidence is needed to demonstrate that a diffusible factor from other cells enables microglia to be infected by the Rubella virus.

We have revised the manuscript to indicate that our data is consistent with the possibility that a diffusible factor is involved. Our experiment utilizing transwell assay argues against phagocytosis and physical interactions as primary drivers, but future studies will be needed to determine if soluble factors are involved.

The authors did not detect Rubella virus transcripts in the single-cell RNA sequencing experiment, nor was a microglia cluster found.

Indeed, microglia recovery using scRNAseq is very inefficient. We note this limitation in the discussion.

Innate immune responses can be activated in the presence of viral particles but without virus replication, as in inactivated viral vaccines; therefore changes in interferon responses do not necessarily prove virus replication.

We agree with the reviewer on this point, it is difficult, if at all possible, to entirely eliminate the possibility that some of the transcriptomic changes, particularly the interferon responses, are not induced by the exposure to viral particles. We have revised the manuscript to more rigorously described the conditions as “RV-exposed”.

Figure 4: it would be helpful to define the abbreviations used in the figure legend (e.g. IPC, RG, EN). In the volcano plots, the gene names are blocked by the dots, and the figure becomes very pixelated when enlarged to read the text.

We have added abbreviations and replaced the figure files with higher resolution images (Figure 6 in current manuscript).

The value of including Supplemental Figure 2 (MOG) is not clear because it receives little mention in the text and also seems to be previously published data that could be cited.

We have removed the figure and replaced it with a citation and a link to the Cell Browser.

Supplemental Figure 4: In panel G, the legend shows "YH10" and "13325". These terms are not described in the Figure legend, nor did a search of the manuscript identify these terms. In its current form Supp. Fig. 4G is not interpretable. In addition, would be more clear to use the term "RV-infected" instead of "treated" to describe the addition of the virus.

We have expanded the Methods section to include the description of different organoid lines and added a revised legend for Supplementary Figure 4. We do not provide evidence of RV infecting organoids without microglia, therefore we have revised the claims that organoid cells become infected with the virus and replaced it with “RV-exposed” to better reflect the conditions studied.

**Reviewer #3 (Recommendations for the authors):**

Demonstrate and quantify virus replication to provide data to complement the imaging. In order of data quality, plaque assays would be most convincing in demonstrating infection and release of infectious virus, while a time course of PCR on RV transcripts would support a conclusion of replicating virus. Further, staining with an anti-double-stranded RNA antibody (J2) would represent evidence of virus replication.

In response to the reviewer’s comment, we performed an RV titering experiment in dissociated microglia co-cultured with other cell types, as well as Vero cells control. While we can detect a robust increase in viral titer from Vero cells, it fell below detection levels in microglia co-cultures. We now include these data in Supplementary Figure 2D.

**Author response image 16. sa3fig16:** Rubella virus titering experiment performed in Vero cells (positive control) or dissociated microglia co-cultures. In primary microglia co- cultures, viral titer falls below detection levels after several days of infection.

We detected a very modest increase in RV RNA in infected brain slices over time using RT- qPCR (see Author response image 17, not included in current manuscript)

**Author response image 17. sa3fig17:** Modest increase in RV RNA over time in brain slice infections. Rubella virus RNA measured by qPCR relative to GAPDH gene, in n=3 samples (2 technical replicates each condition). Brain slices were exposed to RV, then collected at end of inoculation (4 hours post infection), or at 3 or 5 days post infection, and processed for RNA extraction and RT-qPCR.

Unfortunately, we did not find J2 staining informative because we could detect signal in both wild type RV infection conditions and in heat-inactivated RV, presumably due to native dsRNA species present in cells. We did not detect any increase of difference in the pattern of staining between RV and heat-inactivated virus-exposed conditions (Author response image 18; not included in the manuscript).

**Author response image 18. sa3fig18:** J2 antibody labels dsRNA in both RV-exposed and control heat- inactivated virus conditions, presumably due to native dsRNA that is not unique to the viral replication.

We utilized FISH to detect negative-stranded (non-genomic) RV RNA as an alternative to J2 to indicate RNA replication. However, it proved to be not very sensitive, as a small quantity of negative-strand RV RNA could be detected in highly infected Vero cells, but negative-strand RV RNA was not detected in more modestly infected microglia (based on positive-strand RV RNA quantification), as in Author response image 19, not included in current manuscript.

**Author response image 19. sa3fig19:** FISH probes to positive strand (genomic) and negative strand (replication template) RV RNA in Vero cells and microglia co-cultures. (**A**) Representative images of Vero cells infected with RV (top row) or Zika virus as control (bottom row). At 72hpi, cells were fixed and processed for immunofluorescence with anti-RV capsid antibody (RVcap) or Zika virus antibody (Zika4G2), and then FISH was performed using probes to positive strand (+) or negative strand (-) RV RNA. Negative strand RV RNA difficult to visualize at low-power magnification, and required quantification within cell borders defined by wheat germ agglutinin staining with results in panel B. (**B**) In Vero cells, negative strand RV RNA is detected in strongly infected cells. Infection strength determined by intensity of RV capsid immunofluorescence staining and positive strand RV RNA (RVcap/(+) 2/3 indicates robust infection, RVcap/(+) 1 indicates weak infection). ZIKVinf = Zika virus infected control. (**C**) In microglia co-cultures, positive strand RV RNA detected in cells with RV capsid immunopositivity (RVcap_pos). RVinf = RV infected. RVHI = heat-inactivated RV. (**D**) In microglia co-cultures, negative strand RV RNA quantification not significantly different between mock, heat-inactivated RV (RVHI), or RV- infected conditions (RVinf), including cells with weak positive-strand RV RNA (RVinf, (+)<8) or cells with stronger positive-strand RV RNA (RVinf, (+)>=8). Two biological replicates (bHR60 and bHR61), n indicates number of cells counted.

While we could not detect an increase in the viral particles from microglia mixed cultures, we confirmed the presence of GFP from the RV-GFP reporter construct, and we believe it serves as a proof that the virus can infect microglia cells and lead to production of functional viral protein (see Author response image 20, Figure 1E-F of the current manuscript)

**
Author response image 20.
 sa3fig20:** 

Thus, overall we detect replication of viral RNA and protein (qPCR, RV-GFP), but not an appreciable increase in released newly-made virions. The discussion now reflects this more clearly in the current manuscript.

The claim of requiring a diffusible factor to enable RV infection requires additional data. A suggestion would be to include further characterization of affinity-purified cells to define the levels of cell enrichment and to determine which other cell types are present, It is also important to test the RV infection of the fractionated cell types alone before adding to the microglia, in order to demonstrate whether RV is replicating in cell types other than microglia.

We performed quantifications of RV capsid-positive cells in each of the affinity-purified cell populations: neuron-enriched (purified with PSA-NCAM beads), glia-enriched (PSA-NCAM depleted cell fraction), or non-microglia fraction (“Flow through”, depleted of CD11b+ cells). We show that across each condition, we have low infectivity (ranging from ~1 to 4% of total cell population) after 72 hours post-infection. We include these data in Supplementary Figure 3.

**
Author response image 21.
 sa3fig21:** Rubella infection in non-microglia cells. (**A**) Representative images of different cell types depleted of microglia. Cell cultures were stained RV capsid (green) and DAPI. (**B**) Quantification of total cells that are positive for RV capsid across conditions. (**C**) Quantification of RV+ cells that are not microglia across different cell populations. No statistically significant difference was detected in RV infectivity in cells c-cultured with or without microglia.

Another approach to limit cell heterogeneity would be to use iPSC-derived cells, which are highly enriched as a single cell type as a specific cell type, to test the requirement for additional cell types to achieve RV infection of microglia.

In our prior publication (Popova et al. 2021) we have identified a number of molecular differences between primary and iPSC derived microglia. iPSC derived microglia like cells could show differences in infection tropism from primary microglia, and those results may be difficult to interpret biologically. We agree with the reviewer that iPSC derived cells would be an interesting model, there are now several distinct protocols for deriving microglia like cells from pluripotent stem cells and we feel that embarking on a protocol comparison project would fall outside the scope of the current manuscript.

Consider a longer organoid infection. The authors did not identify viral RNA transcripts in their organoid scRNAseq data after a 72-hour infection. Although the 72-hour time point seems right for cells in 2D culture, it’s possible that the infection in the organoids is slower because the virus has to spread inwardly. It would be worth trying a time course out to 2 weeks, collecting organoids every few days and then imaging and doing pcr or plaque assays. Zoomed-out views that show immunofluorescence of the entire organoid would also be beneficial in assessing organoid quality and immunofluorescent staining to identify cell types.

We performed longer RV infection for two weeks and now present data on RV capsid in microglia in 72 hrs and 2 weeks post-infection (Author response image 22 , Figure 4C of the current manuscript). We have also validated one of the scRNAseq-generated gene candidates in combination with different cell type markers and present data on whole organoids immunostained with NeuN for neurons and EOMES for intermediate progenitor cells that demonstrate the overall structure of the organoids (Author response image 23; Figure 6 of the current manuscript).

**Author response image 23. sa3fig23:** Organoids labeled with splice regulator NOVA1 (magenta), neuronal marker NeuN (green) and intermediate progenitor cell marker EOMES (cyan).

**Author response image 22. sa3fig22:** Microglia in organoids co-localize with RV capsid staining. Organoid with microglia were exposed to RV for 72 hrs or two weeks.